# Online Learning in MDPs with Linear Function Approximation and Bandit Feedback

**Gergely Neu**
Universitat Pompeu Fabra
Barcelona, Spain
`gergely.neu@gmail.com`

**Julia Olkhovskaya**
Universitat Pompeu Fabra
Barcelona, Spain
`julia.olkhovskaya@gmail.com`

## Abstract

We consider the problem of online learning in an episodic Markov decision process, where the reward function is allowed to change between episodes in an adversarial manner and the learner only observes the rewards associated with its actions. We assume that rewards and the transition function can be represented as linear functions in terms of a known low-dimensional feature map, which allows us to consider the setting where the state space is arbitrarily large. We also assume that the learner has a perfect knowledge of the MDP dynamics. Our main contribution is developing an algorithm whose expected regret after $T$ episodes is bounded by $\widetilde{\mathcal{O}}\big(\sqrt{dHT}\big)$, where $H$ is the number of steps in each episode and $d$ is the dimensionality of the feature map.

## 1 Introduction

We study the problem of online learning in episodic Markov Decision Processes (MDP), modelling a sequential decision making problem where the interaction between a learner and its environment is divided into $T$ episodes of fixed length $H$. At each time step of the episode, the learner observes the current state of the environment, chooses one of the $K$ available actions, and earns a reward. Consequently, the state of the environment changes according to the transition function of the underlying MDP, as a function of the previous state and the action taken by the learner. A key distinguishing feature of our setting is that we assume that the reward function can change arbitrarily between episodes, and the learner only has access to bandit feedback: instead of being able to observe the reward function at the end of the episode, the learner only gets to observe the rewards that it actually received. As traditional in this line of work, we aim to design algorithms for the learner with theoretical guarantees on her regret, which is the difference between the total reward accumulated by the learner and the total reward of the best stationary policy fixed in hindsight.

Unlike most previous work on this problem, we allow the state space to be very large and aim to prove performance guarantees that do not depend on the size of the state space, bringing theory one step closer to practical scenarios where assuming finite state spaces is unrealistic. To address the challenge of learning in large state spaces, we adopt the classic RL technique of using *linear function approximation* and suppose that we have access to a relatively low-dimensional feature map that can be used to represent policies and value functions. We will assume that the feature map is expressive enough so that all action-value functions can be expressed as linear functions of the features, and that the learner has full knowledge of the transition function of the MDP.

Our main contribution is designing a computationally efficient algorithm called ONLINE Q-REPS, and prove that in the setting described above, its regret is at most $\mathcal{O}\big(\sqrt{dHTD\left(\mu^*\|\mu_0\right)}\big)$, where $d$ is the dimensionality of the feature map and $D\left(\mu^*\|\mu_0\right)$ is the relative entropy between the state-action distribution $\mu^*$ induced by the optimal policy and an initial distribution $\mu_0$ given as input to the

35th Conference on Neural Information Processing Systems (NeurIPS 2021).

algorithm. Notably, our results do not require the likelihood ratio between these distributions to be uniformly bounded, and the bound shows no dependence on the eigenvalues of the feature covariance matrices. Our algorithm itself requires solving a $d^2$-dimensional convex optimization problem at the beginning of each episode, which can be solved to arbitrary precision $\varepsilon$ in time polynomial in $d$ and $1/\varepsilon$, independently of the size of the state-action space.

Our work fits into a long line of research considering online learning in Markov decision processes. The problem of regret minimization in stationary MDPs with a *fixed* reward function has been studied extensively since the work of Burnetas and Katehakis [6], Auer and Ortner [2], Tewari and Bartlett [31], Jaksch et al. [14], with several important advances made in the past decade [9, 10, 4, 13, 15]. While most of these works considered small finite state spaces, the same techniques have been very recently extended to accommodate infinite state spaces under the assumption of realizable function approximation by Jin et al. [17] and Yang and Wang [33]. In particular, the notion of *linear MDPs* introduced by Jin et al. [17] has become a standard model for linear function approximation and has been used in several recent works (e.g., 22, 32, 1).

Even more relevant is the line of work considering adversarial rewards, initiated by Even-Dar et al. [12], who consider online learning in continuing MDPs with full feedback about the rewards. They proposed a MDP-E algorithm, that achieves $\mathcal{O}(\tau^2\sqrt{T\log K})$ regret, where $\tau$ is an upper bound on the mixing time of the MDP. Later, Neu et al. [25] proposed an algorithm which guarantees $\widetilde{\mathcal{O}}\big(\sqrt{\tau^3 KT/\alpha}\big)$ regret with bandit feedback, essentially assuming that all states are reachable with probability $\alpha > 0$ under all policies. In our work, we focus on episodic MDPs with a fixed episode length $H$. The setting was first considered in the bandit setting by Neu et al. [23], who proposed an algorithm with a regret bound of $\mathcal{O}(H^2\sqrt{TK}/\alpha)$. Although the number of states does not appear explicitly in the bound, the regret scales at least linearly with the size of the state space $\mathcal{X}$, since $|\mathcal{X}| \leq H/\alpha$. Later work by Zimin and Neu [35], Dick et al. [11] eliminated the dependence on $\alpha$ and proposed an algorithm achieving $\widetilde{\mathcal{O}}\big(\sqrt{TH|\mathcal{X}|K}\big)$ regret. Regret bounds for the full-information case without prior knowledge of the MDP were achieved by Neu et al. [24] and Rosenberg and Mansour [30], of order $\widetilde{\mathcal{O}}(H|\mathcal{X}|K\sqrt{T})$ and $\widetilde{\mathcal{O}}(H|\mathcal{X}|\sqrt{KT})$, respectively. These results were recently extended to handle bandit feedback about the rewards by Jin et al. [16], ultimately resulting in a regret bound of $\widetilde{\mathcal{O}}(H|\mathcal{X}|\sqrt{KT})$.

As apparent from the above discussion, all work on online learning in MDPs with adversarial rewards considers finite state spaces. The only exception we are aware of is the recent work of Cai et al. [7], whose algorithm OPPO is guaranteed to achieve $\widetilde{\mathcal{O}}\big(\sqrt{d^3H^3T}\big)$, assuming that the learner has access to $d$-dimensional features that can perfectly represent all action-value functions. While Cai, Yang, Jin, and Wang [7] remarkably assumed no prior knowledge of the MDP parameters, their guarantees are only achieved in the full-information case. This is to be contrasted with our results that are achieved for the much more restrictive bandit setting, albeit with the stronger assumption of having full knowledge of the underlying MDP, as required by virtually all prior work in the bandit setting, with the exception of Jin et al. [16].

Our results are made possible by a careful combination of recently proposed techniques for contextual bandit problems and optimal control in Markov decision processes. In particular, a core component of our algorithm is a regularized linear programming formulation of optimal control in MDPs due to Bas-Serrano et al. [5], which allows us to reduce the task of computing near-optimal policies in linear MDPs to a low-dimensional convex optimization problem. A similar algorithm design has been previously used for tabular MDPs by Zimin and Neu [35], Dick et al. [11], with the purpose of removing factors of $1/\alpha$ from the previous state-of-the-art bounds of Neu et al. [23]. Analogously to this improvement, our methodology enables us to make strong assumptions on problem-dependent constants like likelihood ratios between $\mu^*$ and $\mu_0$ or eigenvalues of the feature covariance matrices. Another important building block of our method is a version of the recently proposed Matrix Geometric Resampling procedure of Neu and Olkhovskaya [21] that enables us to efficiently estimate the reward functions. Incorporating these estimators in the algorithmic template of Bas-Serrano et al. [5] is far from straightforward and requires several subtle adjustments.

**Notation.** We use $\langle\cdot,\cdot\rangle$ to denote inner products in Euclidean space and by $\|\cdot\|$ we denote the Euclidean norm for vectors and the operator norm for matrices. For a symmetric positive definite matrix $A$, we use $\lambda_{\min}(A)$ to denote its smallest eigenvalue. We write $\mathrm{tr}\,(A)$ for the trace of a matrix $A$ and use $A \succcurlyeq 0$ to denote that an operator $A$ is positive semi-definite, and we use $A \succcurlyeq B$ to denote

$A - B \succeq 0$. For a $d$-dimensional vector $v$, we denote the corresponding $d \times d$ diagonal matrix by diag($v$). For a positive integer $N$, we use $[N]$ to denote the set of positive integers $\{1, 2, \ldots, N\}$. Finally, we will denote the set of all probability distributions over any set $\mathcal{X}$ by $\Delta_{\mathcal{X}}$.

## 2 Preliminaries

An episodic Markovian Decision Process (MDP), denoted by $M = (\mathcal{X}, \mathcal{A}, H, P, r)$ is defined by a state space $\mathcal{X}$, action space $\mathcal{A}$, episode length $H \in \mathbb{Z}_+$, transition function $P : \mathcal{X} \times \mathcal{A} \to \Delta_{\mathcal{X}}$ and a reward function $r : \mathcal{X} \times \mathcal{A} \to [0, 1]$. For convenience, we will assume that both $\mathcal{X}$ and $\mathcal{A}$ are finite sets, although we allow the state space $\mathcal{X}$ to be arbitrarily large. Without significant loss of generality, we will assume that the set of available actions is the same $\mathcal{A}$ in each state, with cardinality $|\mathcal{A}| = K$. Furthermore, without any loss of generality, we will assume that the MDP has a layered structure, satisfying the following conditions:

- The state set $\mathcal{X}$ can be decomposed into $H$ disjoint sets: $\mathcal{X} = \cup_{h=1}^{H} \mathcal{X}_h$,
- $\mathcal{X}_1 = \{x_1\}$ and $\mathcal{X}_H = \{x_H\}$ are singletons,
- transitions are only possible between consecutive layers, that is, for any $x_h \in \mathcal{X}_h$, the distribution $P(\cdot|x, a)$ is supported on $\mathcal{X}_{h+1}$ for all $a$ and $h \in [H - 1]$.

These assumptions are common in the related literature (e.g., 23, 35, 30) and are not essential to our analysis; their primary role is simplifying our notation.

In the present paper, we consider an *online learning* problem where the learner interacts with its environment in a sequence of episodes $t = 1, 2, \ldots, T$, facing a different *reward functions* $r_{t,1}, \ldots r_{t,H+1}$ selected by a (possibly adaptive) adversary at the beginning of each episode $t$. Oblivious to the reward function chosen by the adversary, the learner starts interacting with the MDP in each episode from the initial state $X_{t,1} = x_1$. At each consecutive step $h \in [H - 1]$ within the episode, the learner observes the state $X_{t,h}$, picks an action $A_{t,h}$ and observes the reward $r_{t,h}(X_{t,h}, A_{t,h})$. Then, unless $h = H$, the learner moves to the next state $X_{t,h+1}$, which is generated from the distribution $P(\cdot|X_{t,h}, A_{t,h})$. At the end of step $H$, the episode terminates and a new one begins. The aim of the learner is to select its actions so that the cumulative sum of rewards is as large as possible.

Our algorithm and analysis will make use of the concept of (stationary stochastic) *policies* $\pi : \mathcal{X} \to \Delta_{\mathcal{A}}$. A policy $\pi$ prescribes a behaviour rule to the learner by assigning probability $\pi(a|x)$ to taking action $a$ at state $x$. Let $\tau^{\pi} = ((X_1, A_1), (X_2, A_2), \ldots, (X_H, A_H))$ be a trajectory generated by following the policy $\pi$ through the MDP. Then, for any $x_h \in \mathcal{X}_h, a_h \in \mathcal{A}$ we define the occupancy measure $\mu_h^{\pi}(x, a) = \mathbb{P}_{\pi}[(x, a) \in \tau^{\pi}]$. We will refer to the collection of these distributions across all layers $h$ as the *occupancy measure* induced by $\pi$ and denote it as $\mu^{\pi} = (\mu_1^{\pi}, \mu_2^{\pi}, \ldots, \mu_H^{\pi})$. We will denote the set of all valid occupancy measures by $\mathcal{U}$ and note that this is a convex set, such that for every element $\mu \in \mathcal{U}$ the following set of linear constraints is satisfied:

$$\sum_{a \in \mathcal{A}} \mu_{h+1}(x, a) = \sum_{x', a' \in \mathcal{X}_h \times \mathcal{A}} P(x|x', a')\mu_h(x', a'), \quad \forall x \in \mathcal{X}_{h+1}, h \in [H - 1], \tag{1}$$

as well as $\sum_a \mu_1(x_1, a) = 1$. From every valid occupancy measure $\mu$, a stationary stochastic policy $\pi = \pi_1, \ldots, \pi_{H-1}$ can be derived as $\pi_{\mu,h}(a|x) = \mu_h(x, a)/\sum_{a'} \mu_h(x, a')$. For each $h$, introducing the linear operators $E$ and $P$ through their action on a set state-action distribution $u_h$ as $(E^{\mathsf{T}} u_h)(x) = \sum_{a \in \mathcal{A}} u_h(x, a)$ and $(P_h^{\mathsf{T}} u_h)(x) = \sum_{x', a' \in \mathcal{X}_h, \mathcal{A}} P(x|x', a')u_h(x', a')$, the constraints can be simply written as $E^{\mathsf{T}}\mu_{h+1} = P_h^{\mathsf{T}}\mu_h$ for each $h$. We will use the inner product notation for the sum over the set of states and actions: $\langle \mu_h, r_h \rangle = \sum_{(x,a) \in (\mathcal{X}_h \times \mathcal{A})} \mu_h(x, a)r_{t,h}(x, a)$. Using this notation, we formulate our objective as selecting a sequence of policies $\pi_t$ for each episode $t$ in a way that it minimizes the *total expected regret* defined as

$$\mathfrak{R}_T = \sup_{\pi^*} \sum_{t=1}^{T} \sum_{h=1}^{H} \left( \mathbb{E}_{\pi^*}\left[ r_{t,h}(X_h^*, A_h^*) \right] - \mathbb{E}_{\pi_t}\left[ r_t(X_{t,h}, A_{t,h}) \right] \right) = \sup_{\mu^* \in \mathcal{U}} \sum_{t=1}^{T} \sum_{h=1}^{H} \langle \mu_h^* - \mu_h^{\pi_t}, r_{t,h} \rangle,$$

where the notations $\mathbb{E}_{\pi^*}[\cdot]$ and $\mathbb{E}_{\pi_t}[\cdot]$ emphasize that the state-action trajectories are generated by following policies $\pi^*$ and $\pi_t$, respectively. As the above expression suggests, we can reformulate our online learning problem as an instance of online linear optimization where in each episode $t$, the

learner selects an occupancy measure $\mu_t \in \mathcal{U}$ (with $\mu_t = \mu^{\pi_t}$) and gains reward $\sum_{h=1}^{H} \langle \mu_{t,h}, r_{t,h} \rangle$. Intuitively, the regret measures the gap between the total reward gained by the learner and that of the best stationary policy fixed in hindsight, with full knowledge of the sequence of rewards chosen by the adversary. This performance measure is standard in the related literature on online learning in MDPs, see, for example Neu et al. [23], Zimin and Neu [35], Neu et al. [24], Rosenberg and Mansour [30], Cai et al. [7].

In this paper, we focus on MDPs with potentially enormous state spaces, which makes it difficult to design computationally tractable algorithms with nontrivial guarantees, unless we make some assumptions. We particularly focus on the classic technique of relying on *linear function approximation* and assuming that the reward functions occurring during the learning process can be written as a linear function of a low-dimensional feature map. We specify the form of function approximation and the conditions our analysis requires as follows:

**Assumption 1** (Linear MDP with adversarial rewards). *There exists a feature map $\varphi : \mathcal{X} \times \mathcal{A} \to \mathbb{R}^d$ and a collection of $d$ signed measures $m = (m_1, \ldots, m_d)$ on $\mathcal{X}$, such that for any $(x, a) \in \mathcal{X} \times \mathcal{A}$ the transition function can be written as*

$$P(\cdot | x, a) = \langle m(\cdot), \varphi(x, a) \rangle .$$

*Furthermore, the reward function chosen by the adversary in each episode $t$ can be written as*

$$r_{t,h}(x, a) = \langle \theta_{t,h}, \varphi(x, a) \rangle$$

*for some $\theta_{t,h} \in \mathbb{R}^d$. We assume that the features and the parameter vectors satisfy $\|\varphi(x, a)\| \leq \sigma$ and that the first coordinate $\varphi_1(x, a) = 1$ for all $(x, a) \in \mathcal{X} \times \mathcal{A}$. Also we assume that $\|\theta_{t,h}\| \leq R$.*

Online learning under this assumption, but with a fixed reward function, has received substantial attention in the recent literature, particularly since the work of Jin et al. [17] who popularized the term "Linear MDP" to refer to this class of MDPs. This has quickly become a common assumption for studying reinforcement learning algorithms (Cai et al. [7], Jin et al. [17], Neu and Pike-Burke [22], Agarwal et al. [1]). This is also a special case of *factored linear models* (Yao et al. [34], Pires and Szepesvári [29]).

Linear MDPs come with several attractive properties that allow efficient optimization and learning. In this work, we will exploit the useful property shown by Neu and Pike-Burke [22] and Bas-Serrano et al. [5] that all occupancy measures in a linear MDP can be seen to satisfy a relaxed version of the constraints in Equation (1). Specifically, for all $h$, defining the feature matrix $\Phi_h \in \mathbb{R}^{(\mathcal{X}_h \times \mathcal{A}) \times d}$ with its action on the distribution $u$ as $\Phi_h^\mathsf{T} u = \sum_{x,a \in \mathcal{X}_h, \mathcal{A}} u_h(x, a) \varphi(x, a)$, we define $\mathcal{U}_\Phi$ as the set of state-action distributions $(\mu, u) = ((\mu_1, \ldots, \mu_H), (u_1, \ldots, u_H))$ satisfying the following constraints:

$$E^\mathsf{T} u_{h+1} = P_h^\mathsf{T} \mu_h \quad (\forall h), \qquad \Phi_h^\mathsf{T} u_h = \Phi_h^\mathsf{T} \mu_h \quad (\forall h), \qquad E^\mathsf{T} u_1 = 1. \tag{2}$$

It is easy to see that for all feasible $(\mu, u)$ pairs, $u$ satisfies the original constraints (1) if the MDP satisfies Assumption 1: since the transition operator can be written as $P_h = \Phi_h M_h$ for some matrix $M_h$. In this case, we clearly have

$$E^\mathsf{T} u_{h+1} = P_h^\mathsf{T} \mu_h = M_h^\mathsf{T} \Phi_h^\mathsf{T} \mu_h = M_h^\mathsf{T} \Phi_h^\mathsf{T} u_h = P_h^\mathsf{T} u_h, \tag{3}$$

showing that any feasible $u$ is indeed a valid occupancy measure. Furthermore, due to linearity of the rewards in $\Phi$, we also have $\langle u_h, r_{t,h} \rangle = \langle \mu_h, r_{t,h} \rangle$ for all feasible $(\mu, u) \in \mathcal{U}_\Phi$. While the number of variables and constraints in Equation (2) is still very large, it has been recently shown that approximate linear optimization over this set can be performed tractably [22, 5]. Our own algorithm design described in the next section will heavily build on these recent results.

## 3 Algorithm and main results

This section presents our main contributions: a new efficient algorithm for the setting described above, along with its performance guarantees. Our algorithm design is based on a reduction to online linear optimization, exploiting the structural results established in the previous section. In particular, we will heavily rely on the algorithmic ideas established by Bas-Serrano et al. [5], who proposed an efficient reduction of approximate linear optimization over the high-dimensional set $\mathcal{U}_\Phi$ to a low-dimensional convex optimization problem. Another key component of our algorithm is an efficient estimator of the reward vectors $\theta_{t,h}$ based on the work of Neu and Olkhovskaya [21]. For reasons that we will clarify in Section 4, accommodating these reward estimators into the framework of Bas-Serrano et al. [5] is not straightforward and necessitates some subtle changes.

### 3.1 The policy update rule

Our algorithm is an instantiation of the well-known "Follow the Regularized Leader" (FTRL) template commonly used in the design of modern online learning methods (see, e.g., 26). We will make the following design choices:

- The decision variables will be the vector $(\mu, u) \in \mathbb{R}^{2(\mathcal{X} \times \mathcal{A})}$, with the feasible set $\mathcal{U}_\Phi^2$ defined through the constraints

$$E^\mathsf{T} u_h = P_h^\mathsf{T} \mu_h \quad (\forall h), \qquad \Phi_h^\mathsf{T} \mathrm{diag}(u_h) \Phi_h = \Phi_h^\mathsf{T} \mathrm{diag}(\mu_h) \Phi_h \quad (\forall h). \qquad (4)$$

These latter constraints ensure that the feature covariance matrices under $u$ and $\mu$ will be identical, which is necessary for technical reasons that will be clarified in Section 4. Notice that, due to our assumption that $\varphi_1(x, a) = 1$, we have $\mathcal{U}_\Phi^2 \subseteq \mathcal{U}_\Phi$, so all feasible $u$'s continue to be feasible for the original constraints (1).

- The regularization function will be chosen as $\frac{1}{\eta} D(\mu \| \mu_0) + \frac{1}{\alpha} D_C(u \| \mu_0)$ for some positive regularization parameters $\eta$ and $\alpha$, where $\mu_0$ is the occupancy measure induced by the uniform $\pi_0$ with $\pi_0(a|x) = \frac{1}{K}$ for all $x, a$, and $D$ and $D_C$ are the marginal and conditional relative entropy functions respectively defined as $D(\mu \| \mu_0) = \sum_{h=1}^H D(\mu_h \| \mu_{0,h})$ and $D_C(\mu \| \mu_0) = \sum_{h=1}^H D_C(\mu_h \| \mu_{0,h})$ with

$$D(\mu_h \| \mu_{0,h}) = \sum_{(x,a) \in (\mathcal{X}_h \times \mathcal{A})} \mu_h(x, a) \log \frac{\mu_h(x, a)}{\mu_{0,h}(x, a)}, \quad \text{and}$$

$$D_C(\mu_h \| \mu_{0,h}) = \sum_{(x,a) \in (\mathcal{X}_h \times \mathcal{A})} \mu_h(x, a) \log \frac{\pi_{\mu,h}(a|x)}{\pi_{0,h}(a|x)}.$$

With these choices, the updates of our algorithm in each episode will be given by

$$(\mu_t, u_t) = \arg \max_{(\mu, u) \in \mathcal{U}_\Phi^2} \left\{ \sum_{s=1}^{t-1} \sum_{h=1}^{H-1} \langle \mu_h, \widehat{r}_{s,h} \rangle - \frac{1}{\eta} D(\mu \| \mu_0) - \frac{1}{\alpha} D_C(u \| \mu_0) \right\} \qquad (5)$$

where $\widehat{r}_{t,h} \in \mathbb{R}^{\mathcal{X} \times \mathcal{A}}$ is an estimator of the reward function $r_{t,h}$ that will be defined shortly.

As written above, it is far from obvious if these updates can be calculated efficiently. The following result shows that, despite the apparent intractability of the maximization problem, it is possible to reduce the above problem into a $d^2$-dimensional unconstrained convex optimization problem:

**Proposition 1.** *Define for each $h \in [H-1]$, a matrix $Z_h \in \mathbb{R}^{d \times d}$ and let matrix $Z \in \mathbb{R}^{d \times d(H-1)}$ be defined as $Z = (Z_1, \ldots, Z_{H-1})$. We will write $h(x) = h$, if $x \in \mathcal{X}_h$. Define the Q-function taking values $Q_Z(x, a) = \varphi(x, a)^\mathsf{T} Z_{h(x)} \varphi(x, a)$ and define the value function*

$$V_Z(x) = \frac{1}{\alpha} \log \left( \sum_{a \in A(x)} \pi_0(a|x) e^{\alpha Q_Z(x,a)} \right)$$

*For any $h \in [H-1]$ and for any $x \in \mathcal{X}_h$, $a \in A(x)$, denote $P_{x,a} V_Z = \sum_{x' \in \mathcal{X}_{h(x)+1}} P(x'|x, a) V_Z(x')$ and $\Delta_{t,Z}(x, a) = \sum_{s=1}^{t-1} \widehat{r}_{s,h(x)}(x, a) + P_{x,a} V_Z - Q_Z(x, a)$. Then, the optimal solution of the optimization problem (5) is given as*

$$\widehat{\pi}_{t,h}(a|x) = \pi_0(a|x) e^{\alpha \left( Q_{Z_t^*}(x,a) - V_{Z_t^*}(x) \right)},$$

$$\widehat{\mu}_{t,h}(x, a) \propto \mu_0(x, a) e^{\eta \Delta_{t, Z_t^*}(x,a)},$$

*where $Z_t^* = (Z_{t,1}^*, \ldots, Z_{t,H-1}^*)$ is the minimizer of the convex function*

$$\mathcal{G}_t(Z) = \frac{1}{\eta} \sum_{h=1}^{H-1} \log \left( \sum_{x \in \mathcal{X}_h, a \in A(x)} \mu_0(x, a) e^{\eta \Delta_{t, Z}(x,a)} \right) + V_Z(x_1). \qquad (6)$$

A particular merit of this result is that it gives an explicit formula for the policy $\pi_t$ that induces the optimal occupancy measure $u_t$, and that $\pi_t(a|x)$ can be evaluated straightforwardly as a function of the features $\varphi(x,a)$ and the parameters $Z_t^*$. The proof of the result is based on Lagrangian duality, and mainly follows the proof of Proposition 1 in Bas-Serrano et al. [5], with some subtle differences due to the episodic setting we consider and the appearance of the constraints $\Phi_h^\mathsf{T}\mathrm{diag}(u_h)\Phi_h = \Phi_h^\mathsf{T}\mathrm{diag}(\mu_h)\Phi_h$. The proof is presented in Appendix A.1.

The proposition above inspires a very straightforward implementation that is presented as Algorithm 1. Due to the direct relation with the algorithm of Bas-Serrano et al. [5], we refer to this method as ONLINE Q-REPS, where Q-REPS stands for "Relative Entropy Policy Search with Q-functions". ONLINE Q-REPS adapts the general idea of Q-REPS to the online setting in a similar way as the O-REPS algorithm of Zimin and Neu [35] adapted the Relative Entropy Policy Search method of Peters et al. [28] to regret minimization in tabular MDPs with adversarial rewards. While O-REPS would in principle be still applicable to the large-scale setting we study in this paper and would plausibly achieve similar regret guarantees, its implementation would be nearly impossible due to the lack of the structural properties enjoyed by ONLINE Q-REPS, as established in Proposition 1.

---

**Algorithm 1** ONLINE Q-REPS

---

**Parameters:** $\eta, \alpha > 0$, exploration parameter $\gamma \in (0, 1)$,
**Initialization:** Set $\widehat{\theta}_{1,h} = 0$ for all $h$, compute $Z_1$.
**For** $t = 1, \ldots, T$, **repeat:**

- Draw $Y_t \sim \mathrm{Ber}(\gamma)$,
- **For** $h = 1, \ldots, H$, **do:**
    - Observe $X_{t,h}$ and, for all $a \in \mathcal{A}(X_{t,h})$, set

    $$\pi_{t,h}(a|X_{t,h}) = \pi_{0,h}(a|X_{t,h})e^{\alpha\big(Q_{Z_t}(X_{t,h},a)-V_{Z_t}(X_{t,h})\big)},$$

    - if $Y = 0$, draw $A_{t,h} \sim \pi_{t,h}(\cdot|X_{t,h})$, otherwise draw $A_{t,h} \sim \pi_{0,h}(\cdot|X_{t,h})$,
    - observe the reward $r_{t,h}(X_{t,h}, A_{t,h})$.
- Compute $\widehat{\theta}_{t,1}, \ldots, \widehat{\theta}_{t,H-1}$, $Z_{t+1}$.

---

## 3.2 The reward estimator

We now turn to describing the reward estimators $\widehat{r}_{t,h}$, which will require several further definitions. Specifically, a concept of key importance will be the following *feature covariance matrix*:

$$\Sigma_{t,h} = \mathbb{E}_{\pi_t}\left[\varphi(X_{t,h}, A_{t,h})\varphi(X_{t,h}, A_{t,h})^\mathsf{T}\right].$$

Making sure that $\Sigma_{t,h}$ is invertible, we can define the estimator

$$\widetilde{\theta}_{t,h} = \Sigma_{t,h}^{-1}\varphi(X_{t,h}, A_{t,h})r_{t,h}(X_{t,h}, A_{t,h}). \tag{7}$$

This estimate shares many similarities with the estimates that are broadly used in the literature on adversarial linear bandits [18, 3, 8]. It is easy to see that $\widetilde{\theta}_{t,h}$ is an unbiased estimate of $\theta_{t,h}$:

$$\mathbb{E}_t\left[\widetilde{\theta}_{t,h}\right] = \mathbb{E}_t\left[\Sigma_{t,h}^{-1}\varphi(X_{t,h}, A_{t,h})\varphi(X_{t,h}, , A_{t,h})^\mathsf{T}\theta_{t,h}\right] = \Sigma_{t,h}^{-1}\Sigma_{t,h}\theta_{t,h} = \theta_{t,h}.$$

Unfortunately, exact computation of $\Sigma_{t,h}$ is intractable. To address this issue, we propose a method to directly estimate the inverse of the covariance matrix $\Sigma_{t,h}$ by adapting the Matrix Geometric Resampling method of Neu and Olkhovskaya [21] (which itself is originally inspired by the Geometric Resampling method of 19, 20). Our adaptation has two parameters $\beta > 0$ and $M \in \mathbb{Z}_+$, and generates an estimate of the inverse covariance matrix through the following procedure[1]:

---

[1]The version we present here is a naïve implementation, optimized for readability. We present a more practical variant in Appendix B

---

**Matrix Geometric Resampling**

---

**Input:** simulator of $P$, policy $\widetilde{\pi}_t = (\widetilde{\pi}_{t,1}, \ldots, \widetilde{\pi}_{t,H-1})$.

**For** $i = 1, \ldots, M$, **repeat**:

    1. Simulate a trajectory
       $\tau(i) = \{(X_1(i), A_1(i)), \ldots, (X_{H-1}(i), A_{H-1}(i))\}$, following the
       policy $\widetilde{\pi}_t$ in $P$,

    2. **For** $h = 1, \ldots, H-1$, **repeat**:
       Compute

       (a) $B_{i,h} = \varphi(X_h(i), A_h(i))\varphi(X_h(i), A_h(i))^\intercal$,

       (b) $C_{i,h} = \prod_{j=1}^{i}(I - \beta B_{j,h})$.

**Return** $\widehat{\Sigma}_{t,h}^{+} = \beta I + \beta \sum_{i=1}^{M} C_{i,h}$ for all $h \in [H-1]$.

---

Based on the above procedure, we finally define our estimator as

$$\widehat{\theta}_{t,h} = \widehat{\Sigma}_{t,h}^{+}\varphi(X_{t,h}, A_{t,h})r_{t,h}(X_{t,h}, A_{t,h}).$$

The idea of the estimate is based on the truncation of the Neumann-series expansion of the matrix $\Sigma_{t,h}^{-1}$ at the $M$th order term. Then, for large enough $M$, the matrix $\Sigma_{t,h}^{+}$ is a good estimator of the inverse covariance matrix, which will be quantified formally in the analysis. For more intuition on the estimate, see section 3.2. in Neu and Olkhovskaya [21]. With a careful implementation explained in Appendix B, $\widehat{\theta}_{t,h}$ can be computed in $O(MHKd)$ time, using $M$ calls to the simulator.

### 3.3 The regret bound

We are now ready to state our main result: a bound on the expected regret of ONLINE Q-REPS. During the analysis, we will suppose that all the optimization problems solved by the algorithm are solved up to an additive error of $\varepsilon \geq 0$. Furthermore, we will denote the covariance matrix generated by the uniform policy at layer $h$ as $\Sigma_{0,h}$, and make the following assumption:

**Assumption 2.** *The eigenvalues of $\Sigma_{0,h}$ for all $h$ are lower bounded by $\lambda_{\min} > 0$.*

Our main result is the following guarantee regarding the performance of ONLINE Q-REPS:

**Theorem 1.** *Suppose that the MDP satisfies Assumptions 1 and 2 and $\lambda_{\min} > 0$. Furthermore, suppose that, for all $t$, $Z_t$ satisfies $\mathcal{G}_t(Z_t) \leq \min_Z \mathcal{G}_t(Z) + \varepsilon$ for some $\varepsilon \geq 0$. Then, for $\gamma \in (0,1)$, $M \geq 0$, positive $\eta \leq \frac{1}{\sigma^2 \beta(M+1)H}$ and any positive $\beta \leq \frac{1}{2\sigma^2\sqrt{d(M+1)}}$, the expected regret of ONLINE Q-REPS over $T$ episodes satisfies*

$$\mathfrak{R}_T \leq 2T\sigma RH \cdot \exp\left(-\gamma\beta\lambda_{\min}M\right) + \gamma HT + \eta TH(3 + 5d) + \frac{1}{\eta}D(\mu^*\|\mu_0)$$

$$+ \frac{1}{\alpha}D_C(u^*\|\mu_0) + \sqrt{\alpha\varepsilon}TH(1 + \eta(M+1)^2).$$

*Furthermore, letting $\beta = \frac{1}{2\sigma^2\sqrt{d(M+1)}}$, $M = \left\lceil \frac{\sigma^4 d\log^2(\sqrt{TH}\sigma R)}{\gamma^2\lambda_{\min}^2} \right\rceil$, $\eta = \frac{1}{\sqrt{TdH}}$, $\alpha = \frac{1}{\sqrt{TdH}}$ and $\gamma = \frac{1}{\sqrt{TH}}$ and supposing that $T$ is large enough so that the above constraints on $M, \gamma, \eta$ and $\beta$ are satisfied, we also have*

$$\mathfrak{R}_T = \widetilde{\mathcal{O}}\left(\sqrt{dHT}\left(1 + D(\mu^*\|\mu_0) + D_C(u^*\|\mu_0)\right) + \sqrt{\varepsilon}(TH)^{9/4}d^{5/4}\frac{1}{\lambda_{\min}^4}\right).$$

Thus, when all optimization problems are solved up to precision $\varepsilon = (TH)^{-7/2}d^{-3/2}\lambda_{\min}^8$, the regret of ONLINE Q-REPS is guaranteed to be of $\mathcal{O}\left(\sqrt{dHTD(\mu^*\|\mu_0)}\right)$.

### 3.4 Implementation

While Proposition 1 establishes the form of the ideal policy updates $\pi_t$ through the solution of an unconstrained convex optimization problem, it is not obvious that this optimization problem can be

solved efficiently. Indeed, one immediate challenge in optimizing $\mathcal{G}_t$ is that its gradient takes the form

$$\nabla \mathcal{G}_t(Z) = \sum_{x,a} \widetilde{\mu}_Z(x,a) \left( \varphi(x,a)\varphi(x,a)^{\mathsf{T}} - \sum_{x',a'} P(x'|x,a)\pi_Z(a'|x')\varphi(x',a')\varphi(x',a')^{\mathsf{T}} \right),$$

where $\widetilde{\mu}_Z(x,a) = \frac{\mu_0(x,a)\exp(\eta\Delta_Z(x,a))}{\sum_{x',a'}\mu_0(x',a')\exp(\eta\Delta_Z(x',a'))}$. Sampling from this latter distribution (and thus obtaining unbiased estimators of $\nabla \mathcal{G}_t(Z)$) is problematic due to the intractable normalization constant.

This challenge can be addressed in a variety of ways. First, one can estimate the gradients via weighted importance sampling from the distribution $\widetilde{\mu}_Z$ and using these in a stochastic optimization procedure. This approach has been recently proposed and analyzed for an approximate implementation of REPS by Pacchiano et al. [27], who showed that it results in $\varepsilon$-optimal policy updates given polynomially many samples in $1/\varepsilon$. Alternatively, one can consider an empirical counterpart of the loss function replacing the expectation with respect to $\mu_0$ with an empirical average over a number of i.i.d. samples drawn from the same distribution. The resulting loss function can then be optimized via standard stochastic optimization methods. This approach has been proposed and analyzed by Bas-Serrano et al. [5]. We describe the specifics of this latter approach in Appendix C.

## 4  Analysis

This section gives the proof of Theorem 1 by stating the main technical results as lemmas and putting them together to obtain the final bound. In the first part of the proof, we show the upper bound on the auxiliary regret minimization game with general reward inputs and ideal updates. Then, we relate this quantity to the true expected regret by taking into account the properties of our reward estimates and the optimization errors incurred when calculating the updates. The proofs of all the lemmas are deferred to Appendix A.

We start by defining the idealized updates $(\widehat{\mu}_t, \widehat{u}_t)$ obtained by solving the update steps in Equation (5) exactly, and we let $u_t$ be the occupancy measure induced by policy $\pi_t$ that is based on the near-optimal parameters $Z_t$ satisfying $\mathcal{G}_t(Z_t) \leq \min_Z \mathcal{G}_t(Z) + \varepsilon$. We will also let $\mu_t$ be the occupancy measure resulting from mixing $u_t$ with the exploratory distribution $\mu_0$ and note that $\mu_{t,h} = (1-\gamma)u_{t,h} + \gamma\mu_{t,h}$. Using this notation, we will consider an auxiliary online learning problem with the sequence of reward functions given as $\widehat{r}_{t,h}(x,a) = \langle \varphi(x,a), \widehat{\theta}_{t,h} \rangle$, and study the performance of the idealized sequence $(\widehat{\mu}_t, \widehat{u}_t)$ therein:

$$\widehat{\mathfrak{R}}_T = \sum_{t=1}^{T} \sum_{h=1}^{H-1} \langle \mu_h^* - \widehat{u}_{t,h}, \widehat{r}_{t,h} \rangle.$$

Our first lemma bounds the above quantity:

**Lemma 1.** *Suppose that $\widehat{\theta}_{t,h}$ is such that $\left| \eta \cdot \langle \varphi(x,a), \widehat{\theta}_{t,h} \rangle \right| < 1$ holds for all $x, a$. Then, the auxiliary regret satisfies*

$$\widehat{\mathfrak{R}}_T \leq \eta \sum_{t=1}^{T} \sum_{h=1}^{H-1} \langle \widehat{\mu}_{t,h}, \widehat{r}_{t,h}^2 \rangle + \frac{1}{\eta} D(\mu^* \| \mu_0) + \frac{1}{\alpha} D_C(u^* \| \mu_0).$$

While the proof makes use of a general potential-based argument commonly used for analyzing FTRL-style algorithms, it involves several nontrivial elements exploiting the structural results concerning ONLINE Q-REPS proved in Proposition 1. In particular, these properties enable us to upper bound the potential differences in a particularly simple way. The main term on contributing to the regret $\widehat{\mathfrak{R}}_T$ can be bounded as follows:

**Lemma 2.** *Suppose that $\varphi(X_{t,h}, a)$ is satisfying $\|\varphi(X_{t,h}, a)\|_2 \leq \sigma$ for any $a$, $0 < \beta \leq \frac{1}{2\sigma^2\sqrt{d(M+1)}}$ and $M > 0$. Then for each $t$ and $h$,*

$$\mathbb{E}_t \left[ \langle \widehat{\mu}_{t,h}, \widehat{r}_{t,h}^2 \rangle \right] \leq 3 + 5d + (M+1)^2 \|\widehat{u}_{t,h} - u_{t,h}\|_1.$$

The proof of this claim makes heavy use of the fact that $\langle \widehat{\mu}_{t,h}, \widehat{r}_{t,h}^2 \rangle = \langle \widehat{u}_{t,h}, \widehat{r}_{t,h}^2 \rangle$, which is ensured by the construction of the reward estimator $\widehat{r}_{t,h}$ and the constraints on the feature covariance matrices

in Equation (4). This property is not guaranteed to hold under the first-order constraints (2) used in the previous works of Neu and Pike-Burke [22] and Bas-Serrano et al. [5], which eventually justifies the higher complexity of our algorithm.

It remains to relate the auxiliary regret to the actual regret. The main challenge is accounting for the mismatch between $\mu_t$ and $u_t$, and the bias of $\widehat{r}_t$, denoted as $b_{t,h}(x,a) = \mathbb{E}_t\left[\widehat{r}_{t,h}(x,a)\right] - r_{t,h}(x,a)$. To address these issues, we observe that for any $t, h$, we have

$$\langle \mu_{t,h}, r_{t,h} \rangle = \langle (1-\gamma)u_{t,h} + \gamma\mu_{0,h}, r_{t,h} \rangle = \langle (1-\gamma)\widehat{u}_{t,h} + \gamma\mu_{0,h}, r_{t,h} \rangle + (1-\gamma)\langle u_{t,h} - \widehat{u}_{t,h}, r_{t,h} \rangle$$
$$\geq \mathbb{E}_t\left[\langle (1-\gamma)\widehat{u}_{t,h} + \gamma\mu_{0,h}, \widehat{r}_{t,h} \rangle\right] + \|b_{t,h}\|_\infty + (1-\gamma)\|u_{t,h} - \widehat{u}_{t,h}\|_1 ,$$

where in the last step we used the fact that $\|r_{t,h}\|_\infty \leq 1$. After straightforward algebraic manipulations, this implies that the regret can be bounded as

$$\mathfrak{R}_T \leq (1-\gamma)\mathbb{E}\left[\widehat{\mathfrak{R}}_T\right] + \sum_{t=1}^{T}\sum_{h=1}^{H}\mathbb{E}\left[\gamma\langle \mu_{0,h} - \mu_h^*, r_{t,h} \rangle + \|\widehat{u}_{t,h} - u_{t,h}\|_1 + \|b_{t,h}\|_\infty\right]. \quad (8)$$

In order to proceed, we need to verify the condition $\left|\eta \cdot \langle \varphi(x,a), \widehat{\theta}_{t,h} \rangle\right| < 1$ so that we can apply Lemma 1 to bound $\widehat{\mathfrak{R}}_T$. This is done in the following lemma:

**Lemma 3.** *Suppose that* $\eta \leq \frac{1}{\sigma^2\beta(M+1)H}$. *Then, for all, $t, h$, the reward estimates satisfy* $\eta\|\widehat{r}_{t,h}\|_\infty < 1$.

Proceeding under the condition $\eta(M+1)$, we can apply Lemma 1 to bound the first term on the right-hand side of Equation (8), giving

$$\mathfrak{R}_T \leq \frac{D(\mu^*\|\mu_0)}{\eta} + \frac{D_C(u^*\|\mu_0)}{\alpha} + (3+5d)\eta HT + \gamma HT$$
$$+ \sum_{t,h}\mathbb{E}\left[(\eta(M+1)^2+1)\|\widehat{u}_{t,h} - u_{t,h}\|_1 + \|b_{t,h}\|_\infty\right].$$

It remains to bound the bias of the reward estimators and the effect of the optimization errors that result in the mismatch between $u_t$ and $\widehat{u}_t$. The following lemma shows that this mismatch can be directly controlled as a function of the optimization error:

**Lemma 4.** *The following bound is satisfied for all $t$ and $h$:* $\|\widehat{u}_{t,h} - u_{t,h}\|_1 \leq \sqrt{2\alpha\varepsilon}$.

The final element in the proof is the following lemma that bounds the bias of the estimator:

**Lemma 5.** *For $M \geq 0$, $\beta = \frac{1}{\sigma^2\beta(M+1)H}$, we have* $\|b_{t,h}\|_\infty \leq \sigma R\exp(-\gamma\beta\lambda_{\min}M)$.

Putting these bounds together with the above derivations concludes the proof of Theorem 1.

## 5 Discussion

This paper studies the problem of online learning in MDPs, merging two important lines of work on this problem concerned with linear function approximation [17, 7] and bandit feedback with adversarial rewards [23, 25, 35]. Our results are the first in this setting and not directly comparable with any previous work, although some favorable comparisons can be made with previous results in related settings. In the tabular setting where $d = |\mathcal{X}||\mathcal{A}|$, our bounds exactly recover the minimax optimal guarantees first achieved by the O-REPS algorithm of Zimin and Neu [35]. For realizable linear function approximation, the work closest to ours is that of Cai et al. [7], who prove bounds of order $\sqrt{d^2H^3T}$, which is worse by a factor of $\sqrt{dH}$ than our result. Their setting, however, is not exactly comparable to ours due to the different assumptions about the feedback about the rewards and the knowledge of the transition function.

One particular strength of our work is providing a complete analysis of the propagation of optimization errors incurred while performing the updates. This is indeed a unique contribution in the related literature, where the effect of such errors typically go unaddressed. Specifically, the algorithms of Zimin and Neu [35], Rosenberg and Mansour [30], and Jin et al. [16] are all based on solving convex optimization problems similar to ours, the effect of optimization errors or potential methods for

solving the optimization problems are not discussed at all. That said, we believe that the methods for calculating the updates discussed in Section 3.4 are far from perfect, and more research will be necessary to find truly practical optimization methods to solve this problem.

The most important open question we leave behind concerns the requirement to have full prior knowledge of $P$. In the tabular case, this challenge has been successfully addressed in the adversarial MDP problem recently by Jin et al. [16], whose technique is based on adjusting the constraints (1) with a confidence set over the transition functions, to account for the uncertainty about the dynamics. We find it plausible that a similar extension of ONLINE Q-REPS is possible by incorporating a confidence set for linear MDPs, as has been done in the case of i.i.d. rewards by Neu and Pike-Burke [22]. Nevertheless, the details of such an extension remain highly non-trivial, and we leave the challenge of working them out open for future work.

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
