# A Omitted proofs

## A.1 The proof of Proposition 1

The proof is based on Lagrangian duality: for each $h \in [H-1]$, we introduce a set of multipliers $V_h \in \mathbb{R}^{|X_h|}$ and $Z_h \in \mathbb{R}^{d \times d}$ corresponding to the two sets of constraints connecting $\mu_{t,h}$ and $u_{t,h}$, and $\rho_{t,h}$ for the normalization constraint of $\mu_{t,h}$. Then, we can write the Lagrangian of the constrained optimization problem as

$$
\begin{aligned}
\mathcal{L}(\mu, u; V, Z, \rho) = & \sum_{h=1}^{H-1} \sum_{s=1}^{t-1} \langle \mu_h, \widehat{r}_{s,h} \rangle + \langle Z_h, \Phi_h^\mathsf{T}(\operatorname{diag}(u_h) - \operatorname{diag}(\mu_h))\Phi_h \rangle \\
& + \sum_{h=1}^{H-1} \left( \rho_h(1 - \langle \mu_h, \mathbf{1} \rangle) - \frac{1}{\eta} D(\mu_h \| \mu_{0,h}) - \frac{1}{\alpha} D_C(u_h \| \mu_{0,h}) \right) \\
& + V_1(x_1)(1 - E^\mathsf{T} u_1) + \sum_{h=1}^{H-1} \langle V_{h+1}, P^\mathsf{T} \mu_h - E^\mathsf{T} u_{h+1} \rangle.
\end{aligned}
$$

For any $h \in [H-1]$, for any $x \in \mathcal{X}_h, a \in A(x)$, denote $Q_Z(x,a) = \varphi(x,a)^\mathsf{T} Z_{h(x)} \varphi(x,a)$, $P_{x,a} V_{h+1} = \sum_{x' \in \mathcal{X}_{h+1}} P(x'|x,a) V_{h+1}(x')$ and $\Delta_{t,Z}(x,a) = \sum_{s=1}^{t-1} \widehat{r}_{s,h(x)}(x,a) + P_{x,a} V_{h(x)+1} - Q_Z(x,a)$. The above Lagrangian is strictly concave, so the maximum of $\mathcal{L}(\mu, d; V, Z, \rho)$ can be found by setting the derivatives with respect to its parameters to zero. This gives the following expressions for the choices of $\pi$ and $\mu$:

$$
\pi_{t,h}^*(a|x) = \pi_{0,h}(a|x) e^{\alpha(Q_Z(x,a) - V_h(x))},
$$

$$
\mu_{t,h}^*(x,a) = \mu_0(x,a) e^{\eta(\Delta_{t,Z}(x,a) - \rho_{t,h})},
$$

From the constraint $\sum_{x \in \mathcal{X}_h, a \in A(x)} \mu_{t,h}^*(x,a) = 1$ for all $h$, we get that

$$
\rho_{t,h}^* = \frac{1}{\eta} \log \left( \sum_{x \in \mathcal{X}_h, a \in A(x)} \mu_0(x,a) e^{\eta \Delta_{t,Z}(x,a)} \right)
$$

and from the constraint $\sum_a \pi_t^*(a|x) = 1$, we get

$$
V_h^*(x) = \frac{1}{\alpha} \log \left( \sum_a \pi_0(a|x) e^{\alpha Q_Z(x,a)} \right).
$$

We will further use the notation $V_Z(x) := V_h^*(x)$. Then, by plugging $\pi_{t,h}^*, \mu_{t,h}^*, V_Z(x)$ into the Lagrangian, we get

$$
\mathcal{G}_t(Z) = \mathcal{L}(\mu^*, u^*; V^*, Z, \rho^*) = \frac{1}{\eta} \sum_{h=1}^{H-1} \log \left( \sum_{x \in \mathcal{X}_h, a \in A(x)} \mu_0(x,a) e^{\eta \Delta_{t,Z}(x,a)} \right) + V_Z(x_1).
$$

Then, the solution of the optimization problem can be written as

$$
\max_{\mu, u \in U} \min_{V,Z,\rho} \mathcal{L}(\mu, u; V, Z, \rho) = \min_{V,Z,\rho} \max_{\mu, u \in U} \mathcal{L}(\mu, u; V, Z, \rho) = \min_Z \mathcal{L}(\mu^*, u^*; V^*, Z, \rho^*) = \min_Z \mathcal{G}_t(Z).
$$

This concludes the proof. ∎

## A.2 The proof of Lemma 1

The proof is based on a variation of the FTRL analysis that studies the evolution of the potential function $\Psi_t$ defined for each $t$ as

$$
\Psi_t = \max_{(\mu, u) \in \mathcal{U}_\Phi^2} \left\{ \sum_{s=1}^{t-1} \sum_{h=1}^{H} \langle \mu_h, \widehat{r}_{s,h} \rangle - \frac{1}{\eta} D(\mu \| \mu_0) - \frac{1}{\alpha} D_C(u \| u_0) \right\}.
$$

This definition immediately implies the following bound:

$$\Psi_{T+1} \geq \sum_{s=1}^{T} \sum_{h=1}^{H-1} \langle \mu_h^*, \widehat{r}_{s,h} \rangle - \frac{1}{\eta} D(\mu^* \| \mu_0) - \frac{1}{\alpha} D_C(u^* \| u_0). \tag{9}$$

To proceed, we will heavily exploit the fact that, by Proposition 1, the potential satisfies $\Psi_t = \min_Z \mathcal{G}_t$. Introducing the notation $Z_t^* = \arg\min_Z \mathcal{G}_t(Z)$, we have

$$\Psi_{t+1} - \Psi_t = \mathcal{G}_{t+1}(Z_{t+1}^*) - \mathcal{G}_t(Z_t^*) \leq \mathcal{G}_{t+1}(Z_t^*) - \mathcal{G}_t(Z_t^*)$$

$$= \frac{1}{\eta} \sum_{h=1}^{H-1} \log \frac{\sum_{x \in \mathcal{X}_h, a \in \mathcal{A}} \mu_{0,h}(x,a) \exp\left(\eta\left(\sum_{s=1}^{t} \widehat{r}_{s,h}(x,a) + P_{x,a} V_{Z_t^*} - Q_{Z_t^*}(x,a)\right)\right)}{\sum_{x' \in \mathcal{X}_h, a' \in \mathcal{A}} \mu_{0,h}(x',a') \exp\left(\eta\left(\sum_{s=1}^{t-1} \widehat{r}_{s,h}(x,a) + P_{x',a'} V_{Z_t^*} - Q_{Z_t^*}(x',a')\right)\right)}$$

$$= \frac{1}{\eta} \sum_{h=1}^{H-1} \log \left( \sum_{x \in \mathcal{X}_h, a \in \mathcal{A}} \mu_{t,h}(x,a) \exp\left(\eta \widehat{r}_{t,h}(x,a)\right) \right)$$

(using the expression of $\mu_{t,h}(x,a)$ obtained in Proposition 1)

$$\leq \frac{1}{\eta} \sum_{h=1}^{H-1} \log \left( 1 + \sum_{x \in \mathcal{X}_h, a \in \mathcal{A}} \mu_{t,h}(x,a) \eta \left( \widehat{r}_{t,h}(x,a) + \eta \widehat{r}_{t,h}^2(x,a) \right) \right)$$

$$\leq \sum_{h=1}^{H-1} \left( \langle \mu_{t,h}, \widehat{r}_{t,h} \rangle + \eta \langle \mu_{t,h}, \widehat{r}_t^2 \rangle \right),$$

where in the last two lines we have used the inequalities $e^z \leq 1 + z + z^2$, which holds for $z \leq 1$ and $\log(1+z) \leq z$, which holds for all $z > -1$, which conditions are verified due to our constraint on $\eta$. Summing up both sides for all $t$ and combining the result with the inequality (9), we obtain

$$\widehat{\mathfrak{R}}_T = \sum_{s=1}^{T} \sum_{h=1}^{H-1} \langle \mu_h^*, \widehat{r}_{s,h} \rangle - \sum_{t=1}^{T} \langle \mu_t, \widehat{r}_t \rangle \leq \eta \sum_{t=1}^{T} \sum_{h=1}^{H-1} \langle \mu_{t,h}, \widehat{r}_{t,h}^2 \rangle + \frac{1}{\eta} D(\mu^* \| \mu_0) + \frac{1}{\alpha} D_C(u^* \| u_0),$$

concluding the proof. ∎

### A.3 The proof of Lemma 2

First, we prove the following statement, that would be helpful further in the proof.

**Lemma 6.** *Let $\widetilde{H}$ be a symmetric positive definite matrix, commuting with $\Sigma_{t,h}$, let $V$ be a matrix where the columns are from the orthonormal system of eigenvectors of $\Sigma_{t,h}$, then*

$$tr\left(\widetilde{H} A_k \Sigma_{t,h} A_k\right) \leq \frac{\beta \sigma^2 tr\left(\Sigma_{t,h}\right)}{2} tr\left(\widetilde{H}\Sigma_{t,h}\left((I - \beta\Sigma_{t,h})^k\right)\right)$$

$$+ \frac{\beta \sigma^2 tr\left(\Sigma_{t,h}\right)}{2} tr\left(\widetilde{H}\Sigma_{t,h}\left(V diag\left(\frac{1}{\lambda_j(\Sigma_{t,h})} \exp\left(\beta^2 \sigma^2 tr\left(\Sigma_{t,h}\right) k + 2\beta\lambda_j(\Sigma_{t,h})\right)\right) V^T\right)\right),$$

*for $j = 1, \ldots, d$.*

*Proof.* To simplify the notation, we omit indices $t, h$ in this proof. Let $H$ be a symmetric positive definite matrix, commuting with $\Sigma_{t,h}$, $X$ be a random vector with $\|X\| \leq \sigma$, then we will show the following inequality that holds almost surely:

$$XX^\mathsf{T} H XX^\mathsf{T} \preccurlyeq \sigma^2 tr\left(H\right) XX^\mathsf{T}.$$

To prove this, we first notice that, since $H$ is symmetric positive definite, we can write $H = \sum_{i=1}^{d} \lambda_i v_i v_i^\mathsf{T}$, and thus

$$XX^\mathsf{T} H XX^\mathsf{T} = \sum_{i=1}^{d} \lambda_i XX^\mathsf{T} v_i v_i^\mathsf{T} XX^\mathsf{T}.$$

To proceed, we will fix $i$ and study the corresponding term in the above sum. Fixing an arbitrary vector $a \in \mathbb{R}^d$ and letting $b_i = v_i X^\mathsf{T} a$, we write

$$a^\mathsf{T} X X^\mathsf{T} v_i v_i^\mathsf{T} X X^\mathsf{T} a = a^\mathsf{T} X v_i^\mathsf{T} X X^\mathsf{T} v_i X^\mathsf{T} a = b_i^\mathsf{T} X X^\mathsf{T} b_i \leq \sigma^2 \|b_i\|_2^2 = \sigma^2 a^\mathsf{T} X v_i^\mathsf{T} v_i X^\mathsf{T} a$$
$$= \sigma^2 \left(a^\mathsf{T} X\right)^2$$

where the inequality is Cauchy–Schwartz and we used that $\|v_i\|_2 = 1$. Multiplying by $\lambda_i$ and summing up on both sides, we get

$$a^\mathsf{T} X X^\mathsf{T} H X X^\mathsf{T} a = \sum_{i=1}^{d} \lambda_i a^\mathsf{T} X X^\mathsf{T} v_i v_i^\mathsf{T} X X^\mathsf{T} a$$

$$\leq \sigma^2 \sum_{i=1}^{d} \lambda_i a^\mathsf{T} X v_i^\mathsf{T} v_i X^\mathsf{T} a$$

$$= \sigma^2 \left(a^\mathsf{T} X\right)^2 \mathrm{tr}\left(H\right).$$

Since the inequality holds for arbitrary $a$, this implies that $X X^\mathsf{T} H X X^\mathsf{T} \preccurlyeq \sigma^2 \mathrm{tr}\left(H\right) X X^\mathsf{T}$. Using the above result and the definition of $A_k = A_{k-1}(I - \beta B_k)$, we get

$$\mathrm{tr}\left(\mathbb{E}\left[\Sigma A_k H A_k\right]\right) \leq \mathrm{tr}\left(\mathbb{E}\left[\Sigma A_{k-1} H \left(I - 2\beta\Sigma\right) A_{k-1}\right]\right) + \beta^2 \sigma^2 \mathrm{tr}\left(H\right) \mathrm{tr}\left(\mathbb{E}\left[\Sigma A_{k-1} \Sigma A_{k-1}\right]\right). \tag{10}$$

To proceed, let us introduce some shorthand notations: $\alpha(H) = \beta^2 \sigma^2 \mathrm{tr}\left(H\right)$, $\alpha = \alpha(\Sigma)$, and $U = I - \beta\Sigma$. Thus, we can rewrite (10) as

$$\mathrm{tr}\left(\mathbb{E}\left[\widetilde{H} A_k H A_k\right]\right) \leq \mathrm{tr}\left(\mathbb{E}\left[\widetilde{H} A_{k-1}\left(\alpha(H)\Sigma + HU\right) A_{k-1}\right]\right). \tag{11}$$

We show that the following holds:

$$\mathrm{tr}\left(\mathbb{E}\left[\widetilde{H} A_k \Sigma A_k\right]\right) \leq \mathrm{tr}\left(\widetilde{H}\Sigma\left(U^k + \alpha \cdot \left(\sum_{i=0}^{k-1}(\alpha + 1)^i U^{k-1-i}\right)\right)\right). \tag{12}$$

To prove the inequality above, we show by induction the following:

$$\mathrm{tr}\left(\mathbb{E}\left[\widetilde{H} A_k \Sigma A_k\right]\right) \leq \mathrm{tr}\left(\widetilde{H} A_{k-j}\Sigma\left(U^j + \alpha \cdot \left(\sum_{i=0}^{(j-1)\wedge 0}(\alpha + 1)^i U^{j-1-i}\right) A_{k-j}\right)\right).$$

First, for $j = 0$, the inequality clearly holds as an equality. Now we will show that if the inequality above holds for $j$, then it also holds for $j + 1$:

$$\mathrm{tr}\left(\widetilde{H} A_{k-j}\Sigma\left(U^j + \alpha \cdot \left(\sum_{i=0}^{(j-1)\wedge 0}(\alpha + 1)^i U^{j-1-i}\right) A_{k-j}\right)\right)$$

$$\leq \mathrm{tr}\left(\widetilde{H} A_{k-j-1}\Sigma\left(U^j + \alpha \cdot \left(\sum_{i=0}^{(j-1)\wedge 0}(\alpha + 1)^i U^{j-1-i}\right)\right) U A_{k-j-1}\right)$$

$$+ \alpha\left(U^j + \alpha \cdot \left(\sum_{i=0}^{(j-1)\wedge 0}(\alpha + 1)^i U^{j-1-i}\right)\right) \mathrm{tr}\left(\widetilde{H} A_{k-j-1}\Sigma A_{k-j-1}\right)$$

$$\leq \mathrm{tr}\left(\widetilde{H} A_{k-j-1}\Sigma\left(U^j + \alpha \cdot \left(\sum_{i=0}^{(j-1)\wedge 0}(\alpha + 1)^i U^{j-1-i}\right)\right) U A_{k-j-1}\right)$$

$$+ \left(\alpha + \alpha \cdot \left(\sum_{i=0}^{(j-1)\wedge 0}(\alpha + 1)^i\right)\right) \mathrm{tr}\left(\widetilde{H} A_{k-j-1}\Sigma A_{k-j-1}\right)$$

$$\leq \mathrm{tr}\left(\widetilde{H} A_{k-j-1}\Sigma\left(U^j + \alpha \cdot \left(\sum_{i=0}^{(j-1)\wedge 0}(\alpha + 1)^i U^{j-1-i}\right)\right) U A_{k-j-1}\right)$$

$$+ (\alpha + 1)^j \mathrm{tr}\left( \widetilde{H} A_{k-j-1} \Sigma A_{k-j-1} \right)$$

$$\leq \mathrm{tr}\left( \widetilde{H} A_{k-j-1} \Sigma \left( U^{j+1} + \alpha \cdot \left( \sum_{i=0}^{j} (\alpha+1)^i U^{j-1-i} \right) \right) A_{k-j-1} \right),$$

which is exactly what we wanted to show. Thus, we proved (12). Now, we use the inequality $1 + x \leq e^x$, to bound $\alpha + 1 \leq e^\alpha$ and $\lambda_i(I - 2\beta\Sigma) \leq e^{-2\beta\lambda_j(\Sigma)}, \forall j \in [d]$, where $\lambda_j(\cdot)$ is the $j$th eigenvalue. It gives us

$$\lambda_j\left( \alpha \left( \sum_{i=0}^{k-1} (\alpha+1)^i U^{k-1-i} \right) \right) \leq \alpha \sum_{i=0}^{k-1} e^{\alpha i} \exp(-2\beta\lambda_j(\Sigma)(k-1-i))$$

$$= \alpha \exp(-2\beta\lambda_j(\Sigma)(k-1)) \sum_{i=0}^{k-1} \exp(\alpha i + 2\beta\lambda_j(\Sigma)i)$$

$$= \alpha \exp(-2\beta\lambda_j(\Sigma)(k-1)) \frac{\exp\left( (\alpha + 2\beta\lambda_j(\Sigma)) k \right) - 1}{\exp(\alpha + 2\beta\lambda_j(\Sigma)) - 1}$$

$$\leq \alpha \exp(-2\beta\lambda_j(\Sigma)(k-1)) \frac{\exp\left( (\alpha + 2\beta\lambda_j(\Sigma)) k \right)}{\exp(\alpha + 2\beta\lambda_j(\Sigma)) - 1}$$

$$\leq \alpha \exp(-2\beta\lambda_j(\Sigma)(k-1)) \exp\left( (\alpha + 2\beta\lambda_j(\Sigma)) (k-1) \right) \frac{\exp\left( (\alpha + 2\beta\lambda_j(\Sigma)) \right)}{\exp(\alpha + 2\beta\lambda_j(\Sigma)) - 1}$$

$$\leq \alpha \exp\left( \alpha(k-1) \right) \frac{\exp\left( \alpha + 2\beta\lambda_j(\Sigma) \right)}{\alpha + 2\beta\lambda_j(\Sigma)}$$

$$\leq \frac{\alpha}{2\beta\lambda_j(\Sigma)} \exp\left( \alpha(k-1) \right) \exp\left( \alpha + 2\beta\lambda_j(\Sigma) \right)$$

$$\text{(using } 1 + x \leq e^x \text{ again)}$$

$$= \frac{\beta\sigma^2 \mathrm{tr}(\Sigma)}{2\lambda_j(\Sigma)} \exp\left( \alpha k + 2\beta\lambda_j(\Sigma) \right) = \frac{\beta\sigma^2 \mathrm{tr}(\Sigma)}{2\lambda_j(\Sigma)} \exp\left( \beta^2\sigma^2 \mathrm{tr}(\Sigma) k + 2\beta\lambda_j(\Sigma) \right).$$

The statement of the lemma is obtained by joining the result of last equation with the equation (12). ∎

We start by using the the definition of $\widehat{\theta}_{t,h}$ to obtain

$$\mathbb{E}_t\left[ \sum_{x \in \mathcal{X}_h, a \in \mathcal{A}} \widehat{\mu}_{t,h}(x,a)\langle \varphi(x,a), \widehat{\theta}_{t,h} \rangle^2 \right]$$

$$= \mathbb{E}_t\left[ \sum_{x \in \mathcal{X}_h, a \in \mathcal{A}} \widehat{\mu}_{t,h}(x,a)\mathrm{tr}\left( \varphi(x,a)\varphi(x,a)^\intercal \widehat{\theta}_{t,h}\widehat{\theta}_{t,h}^\intercal \right) \right]$$

$$= \mathbb{E}_t\left[ \sum_{x \in \mathcal{X}_h, a \in \mathcal{A}} \widehat{u}_{t,h}(x,a)\mathrm{tr}\left( \varphi(x,a)\varphi(x,a)^\intercal \widehat{\theta}_{t,h}\widehat{\theta}_{t,h}^\intercal \right) \right]$$

$$\text{(by the constraint } \Phi_h^\intercal \mathrm{diag}(\widehat{\mu}_t)\Phi_h = \Phi_h^\intercal \mathrm{diag}(\widehat{u}_t)\Phi_h)$$

$$= \mathbb{E}_t\left[ \sum_{x \in \mathcal{X}_h, a \in \mathcal{A}} u_{t,h}(x,a)\mathrm{tr}\left( \varphi(x,a)\varphi(x,a)^\intercal \widehat{\theta}_{t,h}\widehat{\theta}_{t,h}^\intercal \right) \right]$$

$$+ \sum_{x \in \mathcal{X}_h, a \in \mathcal{A}} (u_{t,h}(x,a) - \widehat{u}_{t,h}(x,a))\, \mathbb{E}_t\left[ \langle \varphi(x,a), \widehat{\theta}_{t,h} \rangle^2 \right]$$

$$\leq \mathbb{E}_t\left[ \sum_{x \in \mathcal{X}_h, a \in \mathcal{A}} u_{t,h}(x,a)\mathrm{tr}\left( \varphi(x,a)\varphi(x,a)^\intercal \widehat{\theta}_{t,h}\widehat{\theta}_{t,h}^\intercal \right) \right]$$

$$+ \left\| u_{t,h} - \widehat{u}_{t,h} \right\|_1 \cdot \left\| \mathbb{E}_t \left[ \widehat{r}_{t,h}^2 \right] \right\|_\infty .$$

The second term can be bounded straightforwardly by $\left\| u_{t,h} - \widehat{u}_{t,h} \right\|_1 (M+1)^2$, using Lemma 3 to bound $\left\| \widehat{r}_{t,h} \right\|_\infty \le (M+1)$. As for the first term, we have

$$(1-\gamma)\mathbb{E}_t \left[ \sum_{x,a} u_{t,h}(x,a) \mathrm{tr} \left( \varphi(x,a)\varphi(x,a)^\top \widehat{\theta}_{t,h}\widehat{\theta}_{t,h}^\top \right) \right]$$

$$\le (1-\gamma)\mathbb{E}_t \left[ \sum_{x,a} \mathrm{tr} \left( u_{t,h}(x,a)\varphi(x,a)\varphi(x,a)^\top \widehat{\Sigma}_{t,h}^+ \varphi(X_{t,h}, A_{t,h})\varphi(X_{t,h}, A_{t,h})^\top \widehat{\Sigma}_{t,h}^+ \right) \right]$$

$$\le (1-\gamma)\mathbb{E}_t \left[ \sum_{x,a} \mathrm{tr} \left( u_{t,h}(x,a)\varphi(x,a)\varphi(x,a)^\top \widehat{\Sigma}_{t,h}^+ \varphi(X_{t,h}, A_{t,h})\varphi(X_{t,h}, A_{t,h})^\top \widehat{\Sigma}_{t,h}^+ \right) \right]$$

$$+ \gamma\mathbb{E}_t \left[ \sum_{x,a} \mathrm{tr} \left( u(x,a)\varphi(x,a)\varphi(x,a)^\top \widehat{\Sigma}_{t,h}^+ \varphi(X_{t,h}, A_{t,h})\varphi(X_{t,h}, A_{t,h})^\top \widehat{\Sigma}_{t,h}^+ \right) \right]$$

$$= \mathbb{E}_t \left[ \mathrm{tr} \left( \Sigma_{t,h}\widehat{\Sigma}_{t,h}^+ \Sigma_{t,h}\widehat{\Sigma}_{t,h}^+ \right) \right],$$

where we used $|r_{t,h}(X_{t,h}, A_{t,h})| \le 1$ in the first inequality. For ease of readability, we will omit the indices $h$ in the rest of the proof. Using the definition of $\Sigma_t^+$ and elementary manipulations, we get

$$\mathbb{E}_t \left[ \mathrm{tr} \left( \Sigma_t \Sigma_t^+ \Sigma_t \Sigma_t^+ \right) \right] = \beta^2 \cdot \mathbb{E}_t \left[ \mathrm{tr} \left( \Sigma^* \left( \sum_{k=0}^M C_k \right) \Sigma_t \left( \sum_{j=0}^M C_j \right) \right) \right]$$

$$= \beta^2 \mathbb{E}_t \left[ \sum_{k=0}^M \sum_{j=0}^M \mathrm{tr} \left( \Sigma_t C_k \Sigma_t C_j \right) \right]$$

$$= \beta^2 \mathbb{E}_t \left[ \sum_{k=0}^M \mathrm{tr} \left( \Sigma_t C_k \Sigma_t C_k \right) \right] + 2\beta^2 \mathbb{E}_t \left[ \sum_{k=0}^M \sum_{j=k+1}^M \mathrm{tr} \left( \Sigma_t C_k \Sigma_t C_j \right) \right].$$

Let us first address the first term on the right hand side. Applying Lemma 6 with $\widetilde{H} = \Sigma_t$, we get

$$\beta^2 \sum_{k=0}^M \mathrm{tr} \left( \mathbb{E} \left[ \Sigma_t C_k \Sigma_t C_k \right] \right) \le \beta^2 \sum_{k=0}^M \mathrm{tr} \left( \Sigma^2 (I - 2\beta\Sigma)^k \right)$$

$$+ \beta^2 \sum_{j=1}^d \sum_{k=0}^M \lambda_j^2(\Sigma) \frac{\beta\sigma^2 \mathrm{tr}(\Sigma)}{2\lambda_j(\Sigma)} \exp \left( \beta^2 \sigma^2 \mathrm{tr}(\Sigma) k + 2\beta\lambda_j(\Sigma) \right)$$

$$= \beta \mathrm{tr} \left( \Sigma (I - (I - \beta\Sigma)^M) \right)$$

$$+ \beta^3 \frac{\sigma^2 \mathrm{tr}(\Sigma)}{2} \sum_{j=1}^d \lambda_j(\Sigma) \exp(2\beta\lambda_j(\Sigma)) \frac{\exp \left( \beta^2 \sigma^2 \mathrm{tr}(\Sigma)(M+1) \right) - 1}{\exp \left( \beta^2 \sigma^2 \mathrm{tr}(\Sigma) \right) - 1}$$

$$\le \beta \mathrm{tr}(\Sigma) + \beta^3 \frac{\sigma^2 \mathrm{tr}(\Sigma)}{2\beta^2 \sigma^2 \mathrm{tr}(\Sigma)} \sum_{j=1}^d \lambda_j(\Sigma) \exp(2\beta\lambda_j(\Sigma)) \exp \left( \beta^2 \sigma^2 \mathrm{tr}(\Sigma)(M+1) \right)$$

$$\le \beta \mathrm{tr}(\Sigma) + \frac{\beta}{2} \sum_{j=1}^d \lambda_j(\Sigma) \exp(2\beta\lambda_j(\Sigma)) \exp \left( \beta^2 \sigma^2 \mathrm{tr}(\Sigma)(M+1) \right)$$

$$\le \beta\sigma^2 d + \frac{\beta}{2} \sum_{j=1}^d \sigma^2 \exp(2\beta\sigma^2) \exp \left( \beta^2 \sigma^4 d(M+1) \right) \le 3,$$

where we used the condition $\beta \le \frac{1}{2\sigma^2 d\sqrt{(M+1)}} \le \frac{1}{2\sigma^2}$ and the fact that $(I - \beta(2 - \beta\sigma^2)\Sigma)^M \succcurlyeq 0$ by the same condition. We also used an observation that our assumption on the contexts implies $\mathrm{tr}(\Sigma) \le \mathrm{tr}(\sigma^2 I) = \sigma^2 d$, so again by our condition on $\beta$ it implies the final bound.

Moving on to the second term, we first note that for any $j > k$, the conditional expectation of $B_j$ given $B_{\leq k} = (B_1, B_2, \ldots B_k)$ satisfies $\mathbb{E}\left[C_k \mid B_{\leq k}\right] = C_k(I - \beta\Sigma)^{j-k}$ due to conditional independence of all $B_j$ given $B_k$, for $i > k$. We make use of this equality by writing

$$\beta^2 \sum_{k=0}^{M} \sum_{j=k+1}^{M} \mathbb{E}\left[\operatorname{tr}\left(\Sigma_t C_k \Sigma_t C_j\right)\right] = \beta^2 \sum_{k=0}^{M} \mathbb{E}\left[\mathbb{E}\left[\sum_{j=k+1}^{M} \operatorname{tr}\left(\Sigma_t C_k \Sigma_t C_j\right)\,\middle|\, B_{\leq k}\right]\right]$$

$$= \beta^2 \sum_{k=0}^{M} \mathbb{E}\left[\mathbb{E}\left[\sum_{j=k+1}^{M} \operatorname{tr}\left(\Sigma_t C_k \Sigma_t C_j (I - \beta\Sigma_t)^{j-k}\right)\,\middle|\, B_{\leq k}\right]\right]$$

$$= \beta \sum_{k=0}^{M} \mathbb{E}\left[\mathbb{E}\left[\operatorname{tr}\left(\Sigma_t C_k \Sigma_t C_k \Sigma_t^{-1}\left(I - (I - \beta\Sigma_t)^{M-k}\right)\right)\,\middle|\, B_{\leq k}\right]\right]$$

$$\leq \beta \sum_{k=0}^{M} \mathbb{E}\left[\mathbb{E}\left[\operatorname{tr}\left(\Sigma_t C_k \Sigma_t C_k \Sigma_t^{-1}\right)\,\middle|\, B_{\leq k}\right]\right]$$

$$\text{(due to } (I - \beta\Sigma_t)^{M-k} \succcurlyeq 0\text{)}$$

$$= \beta \sum_{k=0}^{M} \mathbb{E}\left[\operatorname{tr}\left(C_k \Sigma C_k\right)\right]$$

$$\leq \beta \sum_{k=0}^{M} \operatorname{tr}\left(\Sigma(I - 2\beta\Sigma)^k\right)$$

$$+ \beta \sum_{j=1}^{d} \sum_{k=0}^{M} \lambda_j(\Sigma) \frac{\beta\sigma^2 \operatorname{tr}(\Sigma)}{2\lambda_j(\Sigma)} \exp\left(\beta^2\sigma^2\operatorname{tr}(\Sigma)\, k + 2\beta\lambda_j(\Sigma)\right)$$

$$\text{(applying Lemma 6 with } \widetilde{H} = I\text{)}$$

$$\leq d + \frac{\beta^2\sigma^2\operatorname{tr}(\Sigma)}{2} \sum_{j=1}^{d} \frac{\exp\left(\beta^2\sigma^2\operatorname{tr}(\Sigma)(M+1) + 2\beta\lambda_j(\Sigma)\right) - 1}{\exp\left(\beta^2\sigma^2\operatorname{tr}(\Sigma) + 2\beta\lambda_j(\Sigma)\right) - 1}$$

$$\leq d + \frac{1}{2} \sum_{j=1}^{d} \exp\left(\beta^2\sigma^2\operatorname{tr}(\Sigma)(M+1) + 2\beta\lambda_j(\Sigma)\right)$$

$$\leq d + \frac{1}{2} \sum_{j=1}^{d} \exp\left(\beta^2\sigma^4 d(M+1) + 2\beta\lambda_j(\Sigma)\right) \leq 5d.$$

The proof of the theorem is concluded by putting everything together. ∎

## A.4 The proof of Lemma 5

We first observe that the bias of $\widehat{\theta}_{t,h}$ can be easily expressed as

$$\mathbb{E}_t\left[\widehat{\theta}_{t,h}\right] = \mathbb{E}_t\left[\widehat{\Sigma}_{t,h}^{+} \varphi(X_{t,h}, A_{t,h})\varphi(X_{t,h}, A_{t,h})^{\mathsf{T}}\theta_{t,h}\right]$$

$$= \mathbb{E}_t\left[\widehat{\Sigma}_{t,h}^{+}\right] \mathbb{E}_t\left[\varphi(X_{t,h}, A_{t,h})\varphi(X_{t,h}, A_{t,h})^{\mathsf{T}}\right]\theta_{t,h}$$

$$= \mathbb{E}_t\left[\widehat{\Sigma}_{t,h}^{+}\right] \Sigma_{t,h}\theta_{t,h} = \theta_{t,h} - (I - \beta\Sigma_{t,h})^{M}\theta_{t,h}.$$

Thus, the bias is bounded as

$$\left|\mathbb{E}_t\left[\varphi(X_{t,h}, a)^{\mathsf{T}}(I - \beta\Sigma_{t,h})^{M}\theta_{t,h}\right]\right| \leq \left\|\varphi(X_{t,h}, a)\right\|_2 \cdot \left\|\theta_{t,h}\right\|_2 \left\|(I - \beta\Sigma_{t,h})^{M}\right\|_{\text{op}}.$$

In order to bound the last factor above, observe that $\Sigma_{t,h} \succcurlyeq \gamma\Sigma_h$ due to the uniform exploration used in the first layer by MDP-LINEXP3, which implies that

$$\left\|(I - \beta\Sigma_{t,h})^{M}\right\|_{\text{op}} \leq (1 - \gamma\beta\lambda_{\min})^{M} \leq \exp\left(-\gamma\beta\lambda_{\min}M\right),$$

where the second inequality uses $1 - z \leq e^{-z}$ that holds for all $z$. This concludes the proof. ∎

## A.5 The proof of Lemma 4

The proof consists of two main components: proving that the conditional relative entropy between $u_t$ and $\widehat{u}_t$ can be bounded in terms of the optimization error $\varepsilon$, and then using this quantity to bound the total variation distance between these occupancy measures. For ease of readability, we state these results as separate lemmas.

We will first need the following statement:

**Lemma 7.** $D_C(\widehat{u}_t \| u_t) \leq \alpha \varepsilon$.

The proof follows along similar lines as the proof of Lemma 1 in Bas-Serrano et al. [5]. To preserve clarity, we delegate its proof to Appendix A.6 below. The second lemma lemma bounds the relative entropy between two occupancy measures in terms of their *conditional* relative entropies:

**Lemma 8.** *For any two occupancy measures $u$ and $u'$ and any $h$, we have*

$$D\left(u_h \| u_h'\right) \leq \sum_{k=1}^{h} D_C(u_k \| u_k').$$

*Proof.* The proof follows from exploiting some basic properties of the relative entropy. Specifically, the result follows from the following chain of inequalities:

$$
\begin{aligned}
D(u_h \| u_h') &= D(E^{\mathsf{T}} u_h \| E^{\mathsf{T}} u_h') + D_C(u_h \| u_h') \\
&\qquad \text{(by the chain rule of the relative entropy)} \\
&= D(P^{\mathsf{T}} u_{h-1} \| P^{\mathsf{T}} u_{h-1}') + D_C(u_h \| u_h') \\
&\qquad \text{(by the fact that $u$ and $u'$ are valid occupancy measures)} \\
&\leq D(u_{h-1} \| u_{h-1}') + D_C(u_h \| u_h') \\
&\qquad \text{(by the data processing inequality)} \\
&\leq \cdots \leq \sum_{k=1}^{h} D_C(u_k \| u_k'),
\end{aligned}
$$

where the last step follows from iterating the same argument for all layers. ∎

Putting the above two lemmas together and using Pinsker's inequality, we obtain

$$\|\widehat{u}_{t,h} - u_{t,h}\|_1 \leq \sqrt{2 D\left(\widehat{u}_{t,h} \big\| u_{t,h}\right)} \leq \sqrt{2 \sum_{k=1}^{h} D_C\left(\widehat{u}_{t,k} \big\| u_{t,k}\right)} \leq \sqrt{2 D_C\left(\widehat{u}_t \big\| u_t\right)} \leq \sqrt{2\alpha\varepsilon},$$

concluding the proof of Lemma 4. ∎

## A.6 The proof of Lemma 7

For the proof, let us introduce the notation $\widetilde{\mu}_{t,h}$ with

$$\widetilde{\mu}_{t,h}(x,a) = \frac{\mu_{0,h}(x,a) e^{\eta \Delta_{t,Z_t}(x,a)}}{\sum_{(x',a') \in (\mathcal{X}_h \times \mathcal{A})} \mu_{0,h}(x,a) e^{\eta \Delta_{t,Z_t}(x,a)}}.$$

and also $\mathcal{G}_{t,h}(Z) = \frac{1}{\eta} \log \left( \sum_{x \in \mathcal{X}_h, a \in A(x)} \mu_0(x,a) e^{\eta \Delta_{t,Z}(x,a)} \right)$ and $Z_t^* = \arg\min_Z \mathcal{G}_t(Z)$. Then, observe that

$$
\begin{aligned}
D(\widehat{\mu}_{t,h} \| \widetilde{\mu}_{t,h}) &= \sum_{x,a \in \mathcal{X}_h \times \mathcal{A}} \widehat{\mu}_{t,h}(x,a) \log \frac{\widehat{\mu}_{t,h}(x,a)}{\widetilde{\mu}_{t,h}(x,a)} \\
&= \eta \left\langle \widehat{\mu}_{t,h}, \Delta_{t,Z_t^*} - \mathcal{G}_{t,h}(Z_t^*) \mathbf{1} - \Delta_{t,Z_t} + \mathcal{G}_{t,h}(Z_t) \mathbf{1} \right\rangle \\
&= \eta \left\langle \widehat{\mu}_{t,h}, P_h V_{Z_t^*} - Q_{Z_t^*} - P_h V_{Z_t} + Q_{Z_t} \right\rangle + \eta \left( \mathcal{G}_{t,h}(Z_t^*) - \mathcal{G}_{t,h}(Z_t) \right) \\
&= \eta \sum_{(x,a) \in (\mathcal{X}_h \times \mathcal{A})} \sum_{x' \in \mathcal{X}_{h+1}} \widehat{\mu}_{t,h}(x,a) P(x'|x,a) (V_{Z_t^*}(x') - V_{Z_t}(x'))
\end{aligned}
$$

$$+ \eta \left( \mathcal{G}_{t,h}(Z_t^*) - \mathcal{G}_{t,h}(Z_t) \right) + \eta \sum_{(x,a) \in (\mathcal{X}_h \times \mathcal{A})} \widehat{\mu}_{t,h}(x,a) \varphi(x,a)^\top (Z_{t,h} - Z_{t,h}^*) \varphi(x,a)$$

$$= \eta \sum_{(x',a') \in (\mathcal{X}_{h+1} \times \mathcal{A})} \widehat{u}_{t,h+1}(x',a') (V_{Z_t^*}(x') - V_{Z_t}(x')) + \eta (\mathcal{G}_{t,h}(Z_t) - \mathcal{G}_{t,h}(Z_t^*)).$$

$$+ \eta \sum_{(x,a) \in (\mathcal{X}_h \times \mathcal{A})} \widehat{u}_{t,h}(x,a) \varphi(x,a)^\top (Z_{t,h} - Z_{t,h}^*) \varphi(x,a).$$

Here, the last equality follows from the fact that $(\widehat{\mu}_t, \widehat{u}_t)$ satisfy the constraints of the optimization problem (5). On the other hand, we have

$$D_C(\widehat{u}_{t,h} \| u_{t,h}) = \sum_{(x,a) \in (\mathcal{X}_h \times \mathcal{A})} \widehat{u}_{t,h}(x,a) \log \frac{\widehat{\pi}_{t,h}(a|x)}{\pi_{t,h}(a|x)}$$

$$= \alpha \sum_{(x,a) \in (\mathcal{X}_h \times \mathcal{A})} \widehat{u}_{t,h}(x,a) \sum_{x' \in \mathcal{X}_{h+1}} P(x'|x,a) (V_{Z_t^*}(x') - V_{Z_t}(x'))$$

$$+ \alpha \sum_{(x,a) \in (\mathcal{X}_h \times \mathcal{A})} \widehat{u}_{t,h}(x,a) \varphi(x,a)^\top (Z_{t,h} - Z_{t,h}^*) \varphi(x,a)$$

$$= \alpha \sum_{(x',a') \in (\mathcal{X}_{h+1} \times \mathcal{A})} \widehat{u}_{t,h+1}(x,a) (V_{Z_t^*}(x') - V_{Z_t}(x'))$$

$$+ \alpha \sum_{(x,a) \in (\mathcal{X}_h \times \mathcal{A})} \widehat{u}_{t,h}(x,a) \varphi(x,a)^\top (Z_{t,h} - Z_{t,h}^*) \varphi(x,a),$$

where the last equality follows from the fact that $\widehat{u}_t$ is a valid occupancy measure, as shown in Equation (3). Putting the two equalities together, we get

$$\frac{D(\widehat{\mu}_{t,h} \| \mu_{t,h})}{\eta} - \frac{D_C(\widehat{u}_{t,h} \| u_{t,h})}{\alpha} = \mathcal{G}_{t,h}(Z_t) - \mathcal{G}_{t,h}(Z_t^*).$$

Then, summing up over all $h$ gives

$$\frac{D(\widehat{\mu}_t \| \mu_t)}{\eta} - \frac{D_C(\widehat{u}_t \| u_t)}{\alpha} = \sum_{h=1}^{H} (\mathcal{G}_{t,h}(Z_t) - \mathcal{G}_{t,h}(Z_t^*)) = \mathcal{G}_t(Z_t) - \mathcal{G}_t(Z_t^*) \leq \varepsilon.$$

Reordering gives the result. ∎

### A.7 The proof of Lemma 3

The claim is proven by the following straightforward calculation:

$$\eta \cdot \left| \langle \varphi(X_{t,h}, a), \widehat{\theta}_t \rangle \right| = \eta \cdot \left| \varphi(X_{t,h}, a)^\top \widehat{\Sigma}_{t,h}^+ \varphi(X_{t,h}, a) \langle \varphi(X_{t,h}, a), \theta_t \rangle \right|$$

$$\leq \eta \left| \varphi(X_{t,h}, a)^\top \widehat{\Sigma}_{t,h}^+ \varphi(X_{t,h}, a) \right| \leq \eta \sigma^2 \left\| \widehat{\Sigma}_{t,h}^+ \right\|_{\mathrm{op}}$$

$$\leq \eta \sigma^2 \beta \left( 1 + \sum_{k=1}^{M} \| C_{k,h} \|_{\mathrm{op}} \right) \leq \eta (M+1)/2,$$

where we used the fact that our choice of $\beta$ ensures $\| C_{k,h} \|_{\mathrm{op}} = \left\| \prod_{j=0}^{k} (I - \beta B_{j,h}) \right\|_{\mathrm{op}} \leq 1$. ∎

## B  Fast Matrix Geometric Resampling

The naïve implementation of the MGR procedure presented in the main text requires $O(MKHd + MHd^2)$ time due to the matrix-matrix multiplications involved. In this section we explain how to compute $\widehat{\theta}_t$ in $O(MKHd)$ time, exploiting the fact that the matrices $\widehat{\Sigma}_{t,h}$ never actually need to be computed, since the algorithm only works with products of the form $\widehat{\Sigma}_{t,h} \varphi(X_{t,h}, A_{t,h})$ for vectors $X_{t,h}, h \in [H]$. This motivates the following procedure:

It is easy to see from the above procedure that each iteration $k$ can be computed using $(K + 1)Hd$ vector-vector multiplications: sampling each action $A_h(k)$ takes $Kd$ time due to having to compute the products $\langle \varphi(X_h(k)), \sum_{s=1}^{t-1} \widehat{\theta}_{s,a,h} \rangle$ for each action $a$, and updating $Y_{k,h}$ can be done by computing the product $\langle Y_{k-1,h}, \varphi(X_h(k)) \rangle$. Overall, this results in a total runtime of order $MKHd$ as promised above.

## C  Implementation by optimizing the empirical loss

This section outlines a possible implementation of the policy update steps based on approximate minimization of an empirical counterpart of the loss function $\mathcal{G}_t$. To this end, we define

$$\mathcal{G}_{t,h}(Z) = \frac{1}{\eta} \log \left( \sum_{x,a} \mu_0(x,a) e^{\eta \Delta_{Z,t,h}(x,a)} \right)$$

and its empirical counterpart that replaces the expectation by an empirical mean over state-action pairs sampled from $\mu_0$. Concretely, for all $h$, we let $(X_h(i), A_h(i))_{i=1}^{N}$ be $N$ independent samples from $\mu_0$ that can be obtained by running policy $\pi_0$ in the transition model $P$. Using these samples, we define

$$\widehat{\mathcal{G}}_{t,h}(Z) = \frac{1}{\eta} \log \left( \sum_{n=1}^{N} e^{\eta \Delta_{Z,t,h}(X_h(i), A_h(i))} \right). \tag{13}$$

This objective function has several desirable properties: it is convex in $Z$, has bounded gradients, and is $(\alpha + \eta)$-smooth. Furthermore, its gradients can be evaluated efficiently in $\mathcal{O}(N)$ time, given that we can efficiently evaluate expectations of the form $\sum_{x'} P(x'|x,a)V(x')$. As a result, it can be optimized up to arbitrary precision $\varepsilon$ in time polynomial in $1/\varepsilon$ and $N$.

The downside of this estimator is that it is potentially biased. Nevertheless, as the following lemma shows, it is well-concentrated around the true objective function, under some reasonable conditions:

**Lemma 9.** *Fix $Z$ and suppose that $|\Delta_Z(x,a)| \le B$ for all $x, a$. Then, with probability at least $1 - \delta$, the following holds:*

$$\left| \widehat{\mathcal{G}}_{t,h}(Z) - \mathcal{G}_{t,h}(Z) \right| \le 56 \sqrt{\frac{\log(1/\delta)}{N}}.$$

This statement is a variant of Theorem 1 from Bas-Serrano et al. [5], with the key difference being that being able to exactly calculate expectations with respect to $P(\cdot|x,a)$ enables us to prove a tighter bound.

*Proof.* Let us start by defining the shorthand notations $\widehat{S}_i = \Delta_{Z,t}(X_h(i), A_h(i))$ and $W = \frac{1}{N} \sum_{i=1}^{N} e^{\eta S_i}$. Furthermore, we define the function

$$f(s_1, s_2, \ldots, s_N) = \frac{1}{N} \sum_{i=1}^{N} e^{\eta s_i}$$

and notice that it satisfies the bounded-differences property

$$f(s_1, s_2, \ldots, s_i, \ldots, s_N) - f(s_1, s_2, \ldots, s_i', \ldots, s_N) = \frac{1}{N}\left(e^{\eta s_i} - e^{\eta s_i'}\right) \le \frac{\eta e^{2\eta B}}{N}.$$

Here, the last step follows from Taylor's theorem that implies that there exists a $\chi \in (0,1)$ such that

$$e^{\eta s_i'} = e^{\eta s_i} + \eta e^{\eta \chi (s_i' - s_i)}$$

holds, so that $e^{\eta s_i'} - e^{\eta s_i} = \eta e^{\eta \chi (s_i' - s_i)} \le \eta e^{2\eta B}$, where we used the assumption that $|s_i - s_i'| \le 2B$ in the last step. Notice that our assumption $\eta B \le 1$ further implies that $e^{2\eta B} \le e^2$. Thus, also noticing that $W = f(S_1, \ldots, S_N)$, we can apply McDiarmid's inequality that to show that the following holds with probability at least $1 - \delta'$:

$$|W - \mathbb{E}\left[W\right]| \le \eta e^2 \sqrt{\frac{\log(2/\delta')}{N}}. \tag{14}$$

Thus, we can write

$$\widehat{\mathcal{G}}_{t,h}(\theta) - \mathcal{G}_{t,h}(\theta) = \frac{1}{\eta}\log\left(W\right) - \frac{1}{\eta}\log\left(\mathbb{E}\left[W\right]\right) = \frac{1}{\eta}\log\left(\frac{W}{\mathbb{E}\left[W\right]}\right)$$

$$= \frac{1}{\eta}\log\left(1 + \frac{W - \mathbb{E}\left[W\right]}{\mathbb{E}\left[W\right]}\right) \le \frac{W - \mathbb{E}\left[W\right]}{\eta\mathbb{E}\left[W\right]} \le e^4 \sqrt{\frac{\log(2/\delta')}{N}},$$

where the last line follows from the inequality $\log(1 + u) \le u$ that holds for $u > -1$ and our assumption on $\eta$ that implies $W \ge e^{-2}$. Similarly, we can show

$$\mathcal{G}_{t,h}(\theta) - \widehat{\mathcal{G}}_{t,h}(\theta) = \frac{1}{\eta}\log\left(1 + \frac{\mathbb{E}\left[W\right] - W}{W}\right) \le \frac{\mathbb{E}\left[W\right] - W}{\eta W} \le e^4 \sqrt{\frac{\log(2/\delta')}{N}}.$$

This concludes the proof. $\blacksquare$