# OpenReview forum: " Online learning in MDPs with linear function approximation and bandit feedback. "
_NeurIPS.cc/2021/Conference — NeurIPS 2021 Poster_

### Official Review · Reviewer_9trB · 2021-06-25

**Rating:** 6
**Confidence:** 3

**Summary:**

The authors study the adversarial linear bandit problem under bandit feedback. When the dynamics is known, the authors propose a computationally efficient algorithm to achieve low regret.

**Limitations And Societal Impact:**

The authors have adequately addressed them.

**Main Review:**

I have read an earlier version of this paper, where the authors assume access to a simulator to conduct matrix geometric resampling without assuming known dynamics. Although assuming knowing the dynamics sounds weaker, I actually like the new version since the assumption is clear and more reasonable -- Actually, I think assuming a simulator in an online problem with adversarial reward sounds super wired. Also I think authors may want to add more discussion to highlight the improvement in the new version, which would be very helpful for reviewers like me.

I have some other comments:

1. When assuming the known dynamics, in principle we can compute the "covariance matrix" explicitly. The authors claim this is hard. They may want to add some explanation to at least convince the reader this is hard, which is not clear to me at first glance. This will also highlight the significance of the resampling procedure.

2. When finding Z_t, a minimization problem (6) needs to be solved. It would be great if the authors can add some intuition in the main text why this can be done in a computationally efficient manner. In particular, since it contains a summation of S terms and S could be very large, why the overall complexity only depends on d?

3. The decision variable used is a pair (mu,u). Why using this pair is superior to only using mu? Since both of them are constrained by some linear constraints, I can't see intuitively this is the case.

**Time Spent Reviewing:**

1

---

> ### Author Response · Authors · 2021-08-09
> **response**
>
> Thank you for appreciating the improvement over the previous version of the paper and your helpful comments! Let us respond to each of your concerns below:
>
> - Computing the covariance matrix: This is a computationally challenging problem due to the interaction between the policy and the stochastic state dynamics. To see this, consider the simplest possible special case of contextual bandits with Gaussian contexts, two actions, and a deterministic policy: here, estimating each entry of the covariance matrix corresponds to calculating an expectation of a truncated Gaussian, which itself is a well-studied classical problem with nontrivial complexity. (See, e.g., "Computationally and Statistically Efficient Truncated Regression" by Daskalakis et al., COLT 2019.) One can of course attempt to estimate the entries by general sampling methods (e.g., MCMC), but these will inevitable fare worse than the matrix geometric resampling method. This is already true in the simple case of multi-armed bandits---see, e.g., the original paper of Neu and Bartok (2013) and in particular the comparison with the work of Poland (2005) therein.
>
> - Solving the minimization problem: Potential methods to do this are described in Section 3.4. In short, these approaches are based on the fact that the sum over state-action pairs in Eq.(6) is actually an expectation with respect to the initial distribution \mu_0 which can be approximated efficiently by a sample average.
>
> - Optimizing over (\mu,u): We agree that this formulation is not very intuitive, but it is crucially important to arrive to the dual optimization problem of Eq.(6) that only has d^2 variables. In particular, the additional variables u give rise to the Q-function that enables a compact representation of the policies and the dual objective. For more details on the usefulness of this linear program, please see "Logistic Q-Learning" by Bas-Serrano et al. (AISTATS 2021) and "Convex Q-Learning" by Lu et al. (ACC 2021).

---

> > ### Comment · Reviewer_9trB · 2021-08-11
> > **The responses are helpful. I have increased the rate accordingly.**
> >
> > The responses are helpful. I have increased the rate accordingly.

---

> > > ### Author Response · Authors · 2021-08-14
> > > **thank you!**
> > >
> > > Thank you for reading our response and updating your score!

---

### Official Review · Reviewer_LP1g · 2021-07-16

**Rating:** 7
**Confidence:** 3

**Summary:**

This paper presents a theoretical approach for online learning in Markov decision processes (MDPs) with time-varying adversarial rewards and infinite, and possibly continuous, state spaces. The proposed approach is based on linear assumptions on the structure of the MDP and approximate solutions of a linear programming problem. Theoretical regret bounds and discussions on the implementation of a tractable algorithmic solution are provided.

**Limitations And Societal Impact:**

Limitations are discussed in Sec. 5. Societal impact is not directly applicable, as this is a purely theoretical work.

**Main Review:**

The paper is mostly well written and presents the solution to a problem relatively unexplored in the literature: online learning in MDPs with adversarial rewards over infinite/continuous state spaces. The presentation of the method is mostly rigorous in its mathematical formulations with clear theoretical justifications. However, the paper still has a few remaining clarity issues, which I elaborate below.

Issues:
- The first half of the introduction lacks citations/references to back up some of the claims (e.g., lines 21 and 24).
- The second half of the introduction (line 41 onwards) could be labelled as a "related work" section to better structure the paper.
- $K$ is introduced without prior definition in line 52.
- The notation $\tilde{O}$ is not defined, though it's commonly applied to indicate suppression of log-factors in the big-$O$ notation.
- In the preliminaries section, the reward function is first introduced as if it was stationary (see line 97), but it is later on considered to be varying across time step and episode.
- It is not totally clear whether the algorithm can achieve a vanishing regret as $T \to \infty$, since this would imply that both $\eta \to 0$ and $\alpha\to 0$, i.e., the optimisation problem in Theorem 1 would have to be solved up to arbitrary precision. Could the authors comment on that?

**Time Spent Reviewing:**

3

---

> ### Author Response · Authors · 2021-08-09
> **response**
>
> Thank you for your positive evaluation of our work and your helpful comments! We will take all of them into account when preparing the final version of the paper.
>
> Regarding your comment about the computational complexity increasing with T: indeed, as discussed in Section 3.4., calculating each update takes poly(T) time. This is not unusual in incremental algorithms where updates cannot be calculated in closed form and we do not see this as a serious limitation. We will nevertheless make this more explicit in the final version.

---

> > ### Comment · Reviewer_LP1g · 2021-08-31
> > **Reply to the authors**
> >
> > Thanks for clarifying these concerns!

---

### Official Review · Reviewer_a6uC · 2021-07-16

**Rating:** 7
**Confidence:** 3

**Summary:**

This paper studies the problem of online learning in a finite horizon MDP with known transition function and adversarially changing rewards at each episode and only bandit feedback from those rewards. In particular, the authors consider the case when the state space is large warranting linear function approximation with the assumption that the MDP is a linear MDP with a known feature map. The proposed method is essentially a follow-the-leader algorithm using estimated rewards, which are found by estimating the parameter of the reward function.

**Limitations And Societal Impact:**

See main review for limitations and suggestions.

**Main Review:**

Overall, I think this is a good paper. It is very well-written and tackles an interesting open problem in the field. The related work and introduction are detailed and comprehensive and do a good job of making the contributions of the paper clear. The assumptions are repeatedly described. The regret bound is interesting and significant in that the authors show that sqrt{dHT} regret is possible in this difficult setting. Although the proposed method combines ideas from several prior works, I do not think that this is necessarily bad and it does not seem to detract from the quality of the result.

As the authors point out, the main limitation is that the results apply only in the case where the transition function is exactly known and it is not clear how this might extend beyond. That being said, I still think this setting is important to understand.

The other limitation is that the regret bound depends on the divergence between the initial and optimal distributions, which I do not think is always needed in tabular cases, so this would be interesting to address in future work. It is pointed out that the bound has no dependence on the minimum eigenvalue, but I actually think this would be preferable to the divergence since the latter is asking for something akin to “good coverage” which is immutable in the problem while the former is asking only for a good model, which could be chosen by a practitioner even without good coverage.

Other questions & suggestions:

While it is not strictly necessary, I think that the result of Theorem 1 would feel more complete if one could see exactly the effect of the optimization error of solving G, for example, in terms of samples as described in Appendix C. Right now there is an epsilon that represents this in the main theorem. However, since a great deal of the motivation is tractability, perhaps a corollary would be helpful even though this would just be a simple calculation.

In Lemma 8 in the appendix, doesn’t one need a uniform convergence guarantee since the goal is to minimize over \hat{G}? The statement looks like it holds only for a single Z, but the proof seems to handle it more generally. Also, when can one expect that Delta is bounded?

Is there anything wrong with simply plugging the optimization problem (5) into an off-the-shelf solver if one were to implement this practically instead of solving the unconstrained optimization problem? It is my understanding that many solves can handle relative entropy objectives and the constraints seem tame enough. There is some discussion already about it potentially being intractable, but in what way is this the case? Too many states/actions? or something inherent about objective/constraints?


**Time Spent Reviewing:**

5

---

> ### Author Response · Authors · 2021-08-09
> **response**
>
> Thank you for your positive evaluation of our work and constructive comments! We respond to each of these below:
>
> - Divergence vs. smallest eigenvalue: This is a great point! We agree that the KL divergence between the initial and optimal state-action distributions may be difficult to interpret and it may be preferable to replace it by factors that only depend on the features rather than the raw state representations. We will discuss this issue in the final version.
>
> - Computation of epsilon-precise updates: As the discussion in Sec 4.1 shows, updates can be computed to any precision level epsilon in polynomial time. We can definitely state this as a corollary in the final version.
>
> - Lemma 8: Indeed, this argument needs to be completed by a uniform-convergence argument over Z, which would require an explicit upper bound on Delta. Unfortunately, we are not currently aware of any strong bounds on this quantity, and can only provide bounds that feature complicated problem-dependent quantities, similarly to the work of Pacchiano et al. [27]. We will nevertheless provide a covering argument that features this upper bound on Delta in the final version, and note that there is room for improvement in terms of understanding when Delta can be bounded more tightly.
>
> - Off-the-shelf solvers: The optimization problem (5) is generally intractable due to the large number of variables and constraints. In contrast, the dual optimization problem is unconstrained and has only d^2 variables, and is more directly amenable to stochastic optimization as discussed in Sections 3.4. and 5.

---

### Official Review · Reviewer_hYYK · 2021-07-20

**Rating:** 6
**Confidence:** 3

**Summary:**

This paper studies online learning in linear MDPs with bandit feedback and adversarial losses, and a known transition kernel. The regret bound against the best policy is of order $\sqrt{dHT}$. The proposed algorithm is an extension of Q-REPS of Bas-Serrano et al. which previously only works for the standard tabular setting. The authors claim that Q-REPS has computational advantages over O-REPS: despite the fact that the state and action space can be arbitrarily large, solving the optimization problem in each step only requires poly(d) computation.

**Limitations And Societal Impact:**

The assumption on $\lambda_{\min}$ and the related statistical or computational issue are not emphasized or discussed in detail. It should be more emphasized in e.g. introduction section.

**Main Review:**

Although the strong assumption on the known transition kernel simplifies the problem a lot, the setting of adversarial bandit linear MDP has not been studied before, and the authors propose an interesting algorithm to tackle it. I like the fact that the authors make the algorithm as transparent as possible by detailing every sub-procedure e.g. Matrix Geometric Resampling and how to solve the convex programming using gradient descent. However, I suspect that there is some mis-presentation in the main result, as I pointed out below. I would like to see authors address it in the rebuttal (either modifying the algorithm, or weakening the claimed bounds).

As I can see, the regret bound should actually scale with $\sqrt{\frac{T}{\lambda_{\min}}}$, but not the $\sqrt{T}$ given in Theorem 1. The reason is that in Lemma 1, it is required that $\eta$ is smaller than the order of $\frac{1}{\beta M}$ (because the magnitude of $\widehat{\theta}$ can be as large as $\beta M$). A similar condition (i.e., $\eta\leq \frac{2}{(M+2)H}$) is also specified in Theorem 1. Then the $\frac{1}{\eta}D(\mu^*\|\mu_0)$ will scale at least with $\beta M$. Now in order to make the first regret term $2T\sigma RH\exp(-\gamma \beta\lambda_{\min}M)$ sublinear, one must pick $M=\Omega\left(\frac{1}{\gamma\beta\lambda_{\min}}\right)$. So the regret term $\beta M$ is at least $\Omega\left(\frac{1}{\gamma \lambda_{\min}}\right)$. Further considering another regret term $\gamma HT$, the regret will scale at least with $\sqrt{\frac{T}{\lambda_{\min}}}$.

In practice the learner has no way to know $\lambda_{\min}$ and set $M$ and $\epsilon$ properly.  The authors should discuss how the algorithm works when $\lambda_{\min}$ is unknown.


========== after the discussion phase ==========

The authors agreed to write their bound in the clearer form of $\sqrt{\frac{T}{\lambda_{\min}}}$, and discuss the difficulties of removing $\lambda_{\min}$ assumption in the MDP setting.  These make the paper more complete, so I slightly raised my score.


**Time Spent Reviewing:**

8

---

> ### Author Response · Authors · 2021-08-09
> **response**
>
> Thank you for appreciating our technical contribution, and for bringing up the issue of the dependence on lambda_min! This is a valid point: under the current specification of the hyperparameters \eta, \alpha, \gamma, and M, it is not possible to avoid the dependence on 1/lambda_min in the regret bound. However, this issue can be addressed by slightly changing the setting of these parameters as done in Theorem 2 of Neu and Olkhovskaya (COLT 2020). Specifically, by setting \gamma = \sqrt{(log (T \sigma^2 R))/(HT)} and \eta = \alpha = 1/\sqrt{TdH log (T \sigma^2 R)}, one can see that for large enough values of T, the constraint \eta < 2/(M+1) will be satisfied due to \eta decaying faster with T than 1/M does. This results in a regret bound of order \sqrt{T log T}, without any explicit appearance of lambda_min. The need for prior knowledge of lambda_min is thus only necessary for choosing M, but even this problem can be circumvented by replacing the 1/lambda_min factor in the definition of M by another log T factor. Both these fixes come at the price of the regret bounds holding for "large enough T", where "large enough" means log T > 1/lambda_min, which is arguably not great, but is not unheard of in the online learning literature.
>
> In defense of our guarantees, we are compelled to point out that similar issues arise in every single paper published in the literature on adversarial linear bandits. Indeed, all previous work we are aware of in this literature uses unbiased estimates of the loss vectors based on inverse covariance matrices, and guaranteeing the property \eta*(loss estimate) < 1/2 requires access to an exploratory distribution with lambda_min bounded away from zero. Designing policies with large lambda_min has been a subject of active research in this field, see, e.g., "Volumetric Spanners: an Efficient Exploration Basis for Learning" by Hazan, Karnin and Mehka (COLT 2014) or "Towards Minimax Policies for Online Linear Optimization with Bandit Feedback" by Bubeck, Cesa-Bianchi and Kakade (COLT 2012). It is currently unclear how to adapt these techniques to large MDPs or linear contextual bandits, where the decision space is much more complex than in a standard linear bandit problem. We believe that answering this question can be an interesting direction for future work, but is well beyond the scope of the present paper.
>
> In the final version of the paper, we will provide a detailed discussion of these issues.

---

> > ### Comment · Reviewer_hYYK · 2021-08-14
> > **suggestion on the bound**
> >
> > Thank you for the response.  The suggestion in the response is to change an anytime $\sqrt{T}$ bound to a $\sqrt{T\log T}$ bound that holds only when $T\geq \exp(1/\lambda_{\min})$. I would instead suggest to just write $\sqrt{\frac{T}{\lambda_{\min}}}$, because it is actually "tighter" (when $T\geq \exp(1/\lambda_{\min})$, it holds that $\sqrt{\frac{T}{\lambda_{\min}}}\leq \sqrt{T\log T}$), and it won't make the reader overlook the role of $\lambda_{\min}$.  Please see if this makes sense.

---

> > > ### Author Response · Authors · 2021-08-14
> > > **thank you for this suggestion**
> > >
> > > Thank you, that suggestion makes a lot of sense! Indeed, it should make the role of lambda_min more clear to the reader. We will update the paper accordingly, and highlight that getting rid of 1/lambda_min in the bounds doesn't seem to be possible in this setting, at least in as much as there's no known way of removing these factors in the more well-studied setting of adversarial linear bandits either.

---

> > > > ### Comment · Reviewer_hYYK · 2021-08-16
> > > > **discussion**
> > > >
> > > > To my knowledge (and as you pointed out in a previous post), in adversarial linear bandit with a fixed action set, 1/lambda_min factor is not necessary based on the result of "Volumetric Spanners: an Efficient Exploration Basis for Learning" and "Towards Minimax Policies for Online Linear Optimization with Bandit Feedback", so I don't quite understand your response.  In adversarial linear bandits, these works use "spanners" to perform exploration while you use uniform distribution over actions.  So a natural question is whether it is possible to use their ideas of spanners to improve your bound in MDPs?

---

> > > > > ### Author Response · Authors · 2021-08-24
> > > > > **lambda_min**
> > > > >
> > > > > Thank you for this comment and apologies if our previous response wasn't clear enough.
> > > > >
> > > > > What we meant is that the bounds featuring factors of 1/lambda_min is not a new thing that is specific to the setting we consider in this paper, and that such factors do appear in known bounds on adversarial linear bandits. The two works we cited (and also "Online linear optimization and adaptive routing") precisely aimed to derive computationally efficient methods for calculating policies with large lambda_min. In particular, they aimed to derive methods for general d-dimensional convex decision sets that guarantee that both the runtimes and the 1/lambda_min parameters scale polynomially with d.
> > > > >
> > > > > It is unclear if it is possible to adapt these methods to linear MDPs where the decision set is much more complicated: our formulation represents occupancy measures as a 2XA-dimensional convex set, so the above methods do not give meaningful results for this case. Thus, designing good exploratory policies with large lambda_min for linear MDPs is very far from being straightforward.
> > > > >
> > > > > We hope that this response makes sense; in case you find it sensible, we will include it in the final version of the paper to clarify the challenges. (After further polishing the writing, that is.)

---

### Decision · Program_Chairs · 2021-09-27

**Decision:**

Accept (Poster)

**Comment:**

Several reviewers increased their scores after the discussion phase.
Overall, we agree that this paper makes solid contribution to the direction of online learning MDPs.
Please do implement the promised revisions, especially those related to the discussion with Reviewer hYYK.